# Nanoscale imaging of CD47 informs how plasma membrane modifications shape apoptotic cell recognition

Samy Dufour [1], Pascale Tacnet-Delorme[1], Jean-Philippe Kleman [1], Oleksandr Glushonkov[1], Nicole Thielens [1], Dominique Bourgeois [1] & Philippe Frachet [1✉]

CD47 recognized by its macrophage receptor SIRPα serves as a "don't eat-me" signal protecting viable cells from phagocytosis. How this is abrogated by apoptosis-induced changes in the plasma membrane, concomitantly with exposure of phosphatidylserine and calreticulin "eat-me" signals, is not well understood. Using STORM imaging and single-particle tracking, we interrogate how the distribution of these molecules on the cell surface correlates with plasma membrane alteration, SIRPα binding, and cell engulfment by macrophages. Apoptosis induces calreticulin clustering into blebs and CD47 mobility. Modulation of integrin affinity impacts CD47 mobility on the plasma membrane but not the SIRPα binding, whereas CD47/SIRPα interaction is suppressed by cholesterol destabilization. SIRPα no longer recognizes CD47 localized on apoptotic blebs. Overall, the data suggest that disorganization of the lipid bilayer at the plasma membrane, by inducing inaccessibility of CD47 possibly due to a conformational change, is central to the phagocytosis process.

[1] Univ. Grenoble Alpes, CNRS, CEA, IBS, F-38000 Grenoble, France. ✉email: philippe.frachet@ibs.fr

Phagocytosis of undesirable cells is a fundamental process in multicellular organisms. Beyond its well-known role in the fight against pathogens, it ensures tissue homeostasis, elimination of apoptotic cells in a tolerogenic way, and immune surveillance of cancer cells[1–3]. The phagocytosis of apoptotic cells, referred to as "efferocytosis", is now extensively studied[4], and has been identified when perturbed as responsible for numerous and widespread pathologies, notably with a particular immune prevalence. It may therefore constitute a relevant target to develop therapeutic strategies in the fields of autoimmunity and cancers.

Efferocytosis is a routine cellular process with many stages managed by phagocytes and especially by macrophages, from sensing and testing to the final uptake decision of the selected prey[5,6]. This last step is achieved by a set of interactions occurring at the phagocyte-prey interface. However, how the target cell is engulfed, shapes the macrophage's response in an inflammatory or anti-inflammatory manner and drives its antigen presentation ability, remain to be studied in detail.

Although numerous molecules have been proposed to be part of "the phagocytic synapse" and act as "eat-me" or "don't eat-me" signals, the exact function of many of them remains to be understood to establish links between molecular dysfunctions and pathologies. The "don't eat-me" CD47 protein has been found to be a critical player for inhibition of the phagocytosis when bound, in trans, to the macrophage receptor SIRPα (Signal Regulatory Protein α). The finding that disruption of the SIRPα-CD47 interaction enhances phagocytosis of viable cells[7–9], contributed to the emergence of experimental targeting of CD47 for applications in cancer immunotherapy[10,11]. The function of CD47, also known as integrin-associated protein (IAP)[12] has been linked to the cell membrane plasticity[13] and modulation of cell adhesion properties. Indeed, CD47 was shown to interact, *in cis*, with β3 integrins such as αvβ3, αIIbβ3, and the β1 integrins α2β1, α4β1 and α5β1[12,14] even though the conditions under which CD47 and integrins are physically and functionally connected need to be clarified. A recent study of Morrissey and collaborators[15] demonstrated that CD47 ligation to SIRPα ensures repositioning of SIRPα on the phagocytic synapse, along with suppression of integrin activation on the phagocyte, provoking an efficient blockade of phagocytosis.

Interestingly, it was proposed that the efficiency of the "eat-me" signal calreticulin (CRT) which is a key ligand for apoptotic cell recognition and also shown to potentially influence apoptotic cell immunogenicity[16], is dependent on CD47 redistribution at the cell surface concomitant with a loss of binding to SIRPα[7,17]. In addition, this could influence phosphatidylserine (PS)–mediated engulfment. Indeed, PS, which is the best-characterized signal for phagocytosis of apoptotic cells, can interact with CRT[18–20]. Moreover, it was demonstrated that PS, which may be constitutively exposed on viable cells, is required but not sufficient for the dead or apoptotic cells to be engulfed[21]. CRT was also found on the surface of various viable cells[7,22,23] without inducing their uptake by phagocytes.

These data highlight the fact that exposing "eat-me" or "don't-eat" motifs at the cell surface is not sufficient to prompt phagocyte engulfment or to inhibit it. A central question regarding efferocytosis is therefore how modifications of the apoptotic cell membrane influence the reorganization, i.e. clustering and mobility of the molecular complexes that are necessary to activate the uptake process. Answering this question is fundamental to improve our understanding of how "eat-me" and "don't-eat-me" signaling molecules act with their potential lateral partners and their receptors to contribute to the functional phagocytic synapse.

Furthermore CD47 and membrane-exposed CRT (ecto-CRT) are extensively targeted in developing anti-tumor approaches[11,16,24–26], the optimization of which now requires a better knowledge of the molecular organization of these molecules at the cell surface.

As of today, little is known regarding the nanoscale localization and dynamics of molecules known to be crucial for the control of cell-cell recognition and phagocytosis. Recently Wang and collaborators[27] analyzed the nanoscale organization of CD47 in red blood cells (RBCs) from young and aged mice, and suggested that such organization could influence thrombospondin-1 (TSP1) binding, and thus the process of removal of aged RBCs.

Here, we used STORM super-resolution microscopy and single-particle-tracking (SPT) experiments to analyze the differential cell membrane distribution and diffusion behavior of PS, ecto-CRT, and CD47 on adherent HeLa cells. We compared viable cells to cells in early apoptotic stages induced by UVB irradiation. Because the size of the phagocytic synapse is in the 100-300 nm range, beyond the diffraction limited of visible light, super-resolution microscopy provides a tool for analyzing the spatial organization and possible co-localization of the involved molecular assemblies.

We show that at the nanoscale PS, CRT and CD47 are not, or weakly, co-localized at the cell surface. The results obtained on CD47 dynamics and SIRPα binding on apoptotic cells were compared to those obtained on viable cells treated to modify membrane components. Our study shows that CD47 distributes into a low and a high-mobility population, the fractions of which depend on the applied treatments. Apoptosis leads to a significant increase in CD47 mobility, which nevertheless appears functionally uncorrelated to the loss of SIRPα recognition. Rather, this loss was found to be strongly dependent on cholesterol remodeling.

## Results

### CD47 distribution on apoptotic cells is independent of CRT clustering

It was originally proposed that the CD47 "don't eat-me" function is altered while CD47 moves away from CRT following the onset of apoptosis, which is accompanied by PS exposure[7]. This correlates with an increase of apoptotic cell engulfment by macrophages hypothetically due to the "eat-me" functions of CRT and PS, and to the loss of CD47 cell surface clustering which could affect its avidity-mediated binding to SIRPα[17]. This prompted us to investigate at the nanoscale, using STORM super-resolution imaging, the localization of CD47, CRT and PS during the remodeling of the apoptotic cell membrane. HeLa cells were used as a model (as described in Fig. S1) well adapted to our super-resolution microscopy approach as these adherent cancer cells express CD47 and expose ectopic CRT, which has been characterized as a pro-phagocytic event at early apoptotic stages[16,19,28]. Unpermeabilized cell samples were labeled for CRT, CD47 and PS using primary anti-CRT and secondary Alexa Fluor 532 (A532)-labeled antibodies, an A647-labeled anti CD47 antibody or by A647-labeled annexin V, respectively. Thanks to these combinations, we generated two-color super-resolution STORM images (Fig. 1) from doubly-labeled CRT/CD47 and CRT/PS samples. These images showed a non-homogeneous distribution of the molecules on the apoptotic cell surface. CRT, which was present on the outside face of viable HeLa cells (Fig. 1a and S2a) was noticeably redistributed on the surface of apoptotic cells and mostly clustered in restricted areas, specifically on apoptotic blebs as shown in Fig. 1a-c, which represent different steps of the morphological modifications of the plasma membrane induced by apoptosis. As a control, a similar pattern was observed for ERp57, another endoplasmic reticulum chaperone known to be co-translocated with CRT on the outer leaflet of the plasma membrane[29–31] (Fig. S3a). Unlike CRT, CD47 detected on viable and apoptotic cells remained homogeneously

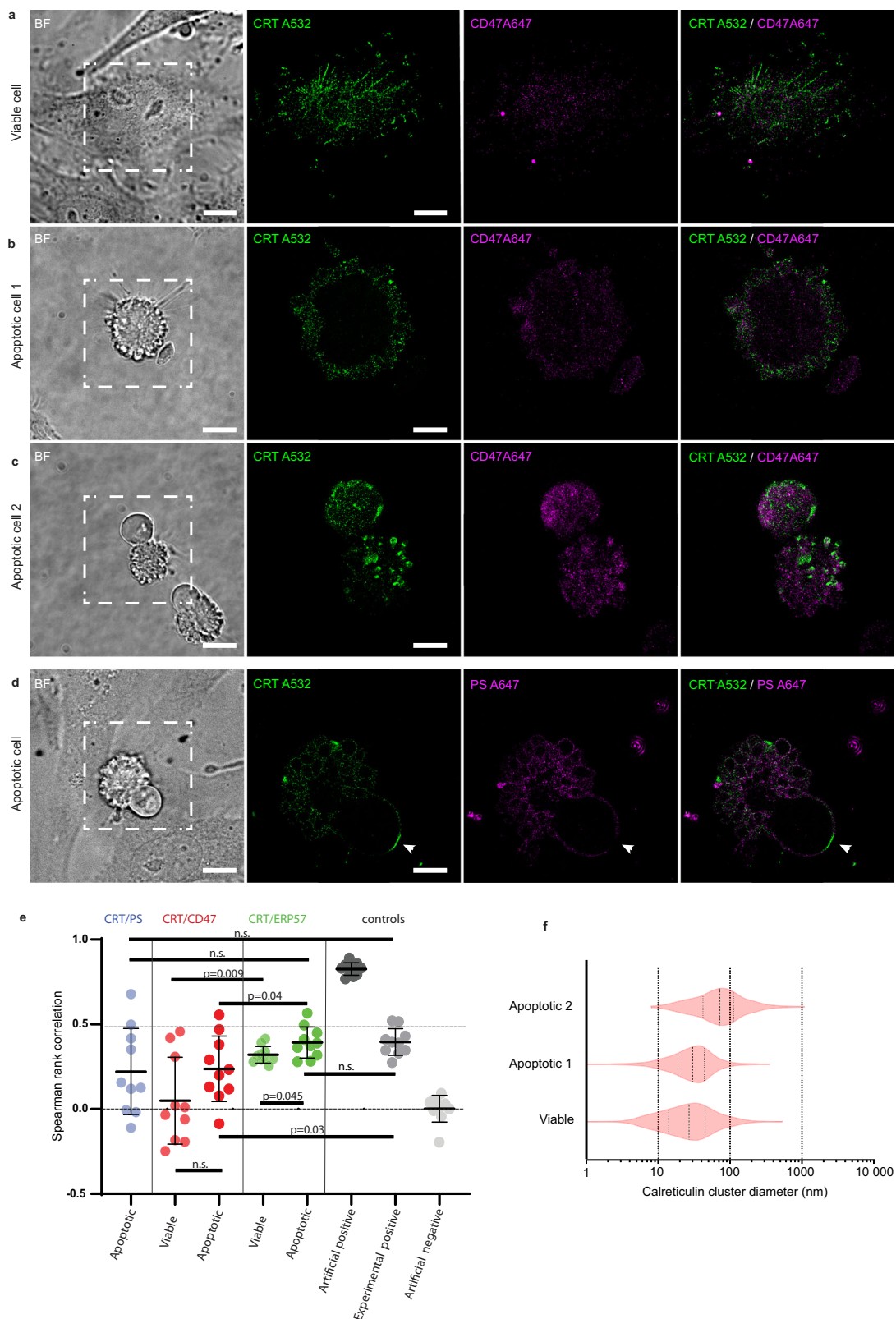

distributed throughout the cell surface (Fig. 1a–c). Although CD47 was found in CRT-enriched regions (Fig. 1a–c), the Spearman's rank correlation analysis statistically indicates a lack of, or a weak, co-localization between CRT and CD47 on viable or apoptotic cells in contrast to what was obtained for CRT/ERp57 which are significantly more co-localized, as confirmed by Spearman's values similar to the positive experimental control (Figs. 1e, S2–5).

Cell labeling with annexin V revealed that PS is distributed all over the cell membrane with a typical decoration of the blebs membrane (Fig. 1d), also enriched with CRT. Nevertheless, analysis at the scale of the entire cell using the Spearman's test indicated a very heterogeneous situation for the correlation between localization of PS and CRT (Fig. 1e). Indeed, a more detailed investigation revealed regions where CRT co-localized

**Fig. 1 Evolution of the nanoscale organization of calreticulin, PS and CD47 during apoptosis.** HeLa cells, viable or 4 h post-UVB-irradiation were stained and imaged by STORM. Bright-field imaging (BF) was used to differentiate healthy cell from apoptotic cell. **a–c** CD47 (magenta) and calreticulin (green) were detected with an anti-CD47 antibody (B6H12) labeled with Alexa Fluor 647 (A647) and with an anti-calreticulin antibody (PA1-902A) revealed by secondary antibodies labeled with Alexa Fluor 532 (A532). Cell 1 (**b**) and cell 2 (**c**) illustrate the clustering of CRT concomitant with the progression of apoptosis. **d** Phosphatidylserine (magenta) and calreticulin (green) were detected with biotinylated annexin V and an anti-calreticulin antibody (PA1-902A) and revealed by streptavidin labeled with A647 and secondary antibodies labeled with A532 fluorophores, respectively. The white arrow points to a specific area where CRT and PS are excluded. Scale bar, 10 μm on BF images and 5 μm on STORM images. **e** 2D co-localization was quantified by Spearman's rank correlation. Ten cells per condition are analyzed from independent experiments ($n \geq 4$). Co-localizations were measured on the whole cells. Exact significant P values (two-sided unpaired t test with Welch's correction) are mentioned as indicated, and n.s. $p > 0.05$. Each dot corresponds to one cell. Means with SD are shown. Experimental, artificial positive and artificial negative controls were obtained as described in the methods section and represented in Fig. S3a, b. **f** Violin plots show representative distribution of CRT cluster diameter measured on the cells represented in Fig. 1a-c. Segmentation and quantification of calreticulin cluster diameter were determined as shown in Fig. S3.c.

with PS (particularly in blebs) and other areas where CRT is present, but PS is excluded (Fig. 1d and S2b). On the one hand these observations are in agreement with published data showing that CRT and PS can interact[7,19,20,32]. On the other hand the fact that CRT can be localized to the membrane in the absence of PS is in line with the observation that CRT exposure can precede PS exposure[18,31] which was also observed in our experimental model (Fig. S6). This highlights the complexity of interactions evolving with the development of apoptosis. In addition, the size of CRT clusters (Fig. 1f and S3c) increases significantly along their re-localization into apoptotic blebs.

Taken together, these results suggest that the distributions of CRT and CD47, their co-localization and concomitant repositioning with PS exposure, cannot be strictly related to their respective functions on the surface of apoptotic cells. In other words, the "don't eat-me" signal CD47 appears to be independent from its co-localization with CRT, while the segregation of CRT away from CD47 seems not required for "eat-me" signaling.

**CD47 displays a low-mobility and a high-mobility form which is enhanced on the apoptotic cell surface.** Our above observation mostly showing a lack of redistribution of CD47 towards CRT during apoptosis, prompted us to examine the dynamical behavior of CD47 at the surface of viable and apoptotic cells. A single particle tracking approach in TIRF mode was thus developed using a low concentration of A647-labeled anti-CD47 antibody (B6H12) (Fig. 2). The density of recorded CD47 tracks, as well as the length of the tracks, were optimized as described in Methods. As a proxy for mobility, we measured the Mean Jump Distance (MJD), that is, the average distance travelled by one particle between two consecutive frames in individual tracks[33]. One advantage of measuring MJDs is that they are independent of an a priori diffusion model. The analysis of the MJD distribution on viable cells revealed two distinct populations, a "low-mobility" (MJDs ≤ 140 nm) and a "high mobility" (MJDs ≥ 140 nm) population (Fig. 2a-2c). On one of our UV-irradiated samples (Fig. 2a), two neighboring cells were spotted, one showing a clear apoptotic phenotype with a characteristic membrane blebbing and the other displaying a more spread and adherent phenotype. The length of CD47 MJDs increased clearly on the cell at a more advanced stage of apoptosis (Fig. 2a). The histogram distribution of MJDs evolved accordingly (Fig. 2b), with an increase of the high mobility sub-population and a decrease of the low mobility population in the cell with a blebbing phenotype, whereas the MJD histogram of the well adherent cell remained similar to that of viable cells. These observations strongly suggest that the CD47 mobility increases upon the onset of apoptosis. Interestingly, analysis of individual tracks reconstructed from viable cells revealed peculiar nanoscale patterns that could not be observed on fixed cells by STORM. This suggests that some CD47 molecules are sequestered in restricted

regions (Fig. 2c) possibly assigned to microvilli and vesicle-like domains, in line with previous experiments[34,35].

**CD47 mobility is dependent on integrin state and on plasma membrane organization.** To test whether the mobility behavior of CD47 is linked to its interaction with other plasma membrane molecules, we performed SPT measurements after treating cells with various chemicals, which impact plasma membrane components. Formaldehyde (FA) fixation is commonly used to fix proteins and preserve the cell cytoskeleton but has a poor ability to immobilize lipids in contrast to glutaraldehyde[36]. Interestingly, after FA fixation, the extent of CD47 tracks diminished (Fig. 3a and b). The MJDs were significantly reduced with the appearance of a fraction of almost immobile molecules. One mobile population of CD47 persisted after FA treatment, but essentially disappeared after glutaraldehyde treatment, which stiffens the membrane, or in the presence of saponin, a detergent known to extract cholesterol, hence triggering destabilization of the lipid rafts (Fig. 3a and b). These observations suggest that the low-mobility CD47 population may be constrained by other macromolecules of the membrane or of the cytoskeleton, while the high-mobility population is diffusing more freely in the lipid bilayer. Changes in CD47 mobility observed in apoptotic cells may thus be controlled by its interaction with plasma membrane components, such as lipids, lipid-associated molecules or proteins that are modified during apoptosis. To test this hypothesis, we decided to treat cells with cytoskeleton, integrins or lipid rafts perturbators. To evaluate how these treatments affect the CD47 mobility, we calculated the value (R) corresponding to the proportion of high-mobility MJDs relative to all MJDs collected from individual cells (Fig. 3c and S7). Thus, a bigger R value is indicative of an increase in CD47 mobility.

**Breaking actin filaments does not affect CD47 mobility.** Cytochalasin D treatment, which disrupts the cytoskeleton by breaking actin filaments, did not affect CD47 mobility (Fig. 3c). This suggests that the change in R observed upon apoptosis is not directly due to an alteration of the cytoskeleton structure but most likely a consequence of a modification of the plasma membrane in link with the depolymerization of actin fibers[37,38].

**Modulation of Integrin affinity state impacts CD47 mobility.** β3 integrins are cell-cell and cell-matrix adhesion receptors whose function (i.e. ligand binding activity) is affected by apoptosis and which are known to be CD47 partners[12]. EDTA treatment, which destabilizes integrin heterodimers and results in a shift toward the inactive forms that abrogate binding to their adhesion ligands[39,40], resulted in a significant decrease of the CD47 R-value (Fig. 3c). In contrast, $Mn^{2+}$, which promotes the high affinity state of integrins, increased the R value, similarly to what we observed on apoptotic cells. In support of an interaction of β3

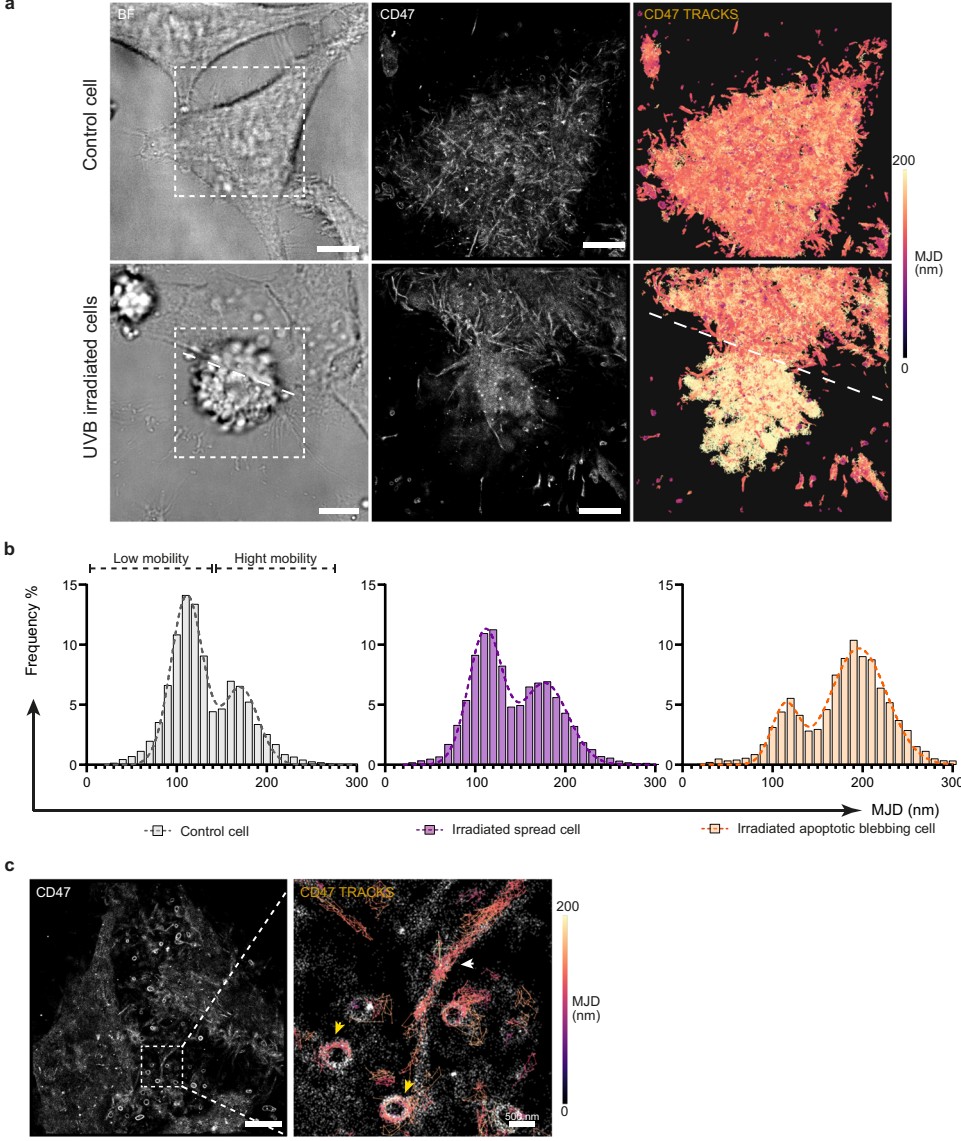

**Fig. 2 Two mobile subpopulations of CD47 coexist on the cell surface and their proportion evolves with apoptosis. a–c** CD47 was detected with an anti-CD47 antibody (B6H12) labeled with A647 and imaged by TIRF microscopy during 120 s at 33 Hz. **a** CD47 was visualized on non-treated (control) or UVB-irradiated HeLa cells. Brightfield images (BF) were used to differentiate spread cells from apoptotic cells (left panels). CD47 localizations were compiled to reconstruct the image (middle panel). Mean Jump Distances (MJDs) were calculated for individual CD47 trajectories and each trajectory was colored according to its MJD value (color scale on the right side). The dotted line in the lower right image delimits the spread cell from the apoptotic cell presenting a blebbing phenotype. Scale bar, 10 μm on BF images, 5 μm on CD47 images. **b** Representative MJD distributions extracted from the acquisition of the cells shown in (**a**). Two classes of CD47 mobility were determined according to their MJD as annotated on the histogram obtained on a viable control cell. **c** Views of CD47 localization from 250 frames tracks showing CD47 sequestered on structures putatively assigned to microvilli (white arrow) and microvesicles (yellow arrows). Scale bar, 5 μm.

integrins with CD47, which could be at the origin of the modified R values, CD47 was efficiently detected from β3 immunoprecipitation on HeLa cells lysate (Fig. 4a). To probe this putative interaction in more detail, we performed two-color CD47/αvβ3 SPT experiments. Strikingly, CD47 and mobile integrins could be detected in the same restricted areas (microvilli or vesicle-like domains, Fig. 4b, selected area 2 and 3) while CD47 did not localize with αvβ3 in areas where integrins appeared immobile (Fig. 4b, selected area 1), expected to be focal adhesion sites. This was confirmed by CD47/vinculin and αvβ3/vinculin double immunolabeling experiments analyzed by confocal microscopy showing that CD47 was not enriched in focal adhesion sites, in contrast to αvβ3 integrin (Fig. S8). This suggests that the low mobility population of CD47 and the mobile population of αvβ3

are constrained inside the same type of membrane sub-domain structure, proposed above as being microvilli or vesicle-like domains (Fig. 2c). In addition, the analysis of the αvβ3 MJDs extracted from SPT acquisitions (Fig. 4c) showed that they are mainly distributed between an immobile population with a mean MJD value around 44 nm (corresponding to the MJD of immobile CD47 molecules determined after glutaraldehyde fixation and shown in Fig. 3b) and a mobile one. This latter is characterized by MJDs close to the mean value determined for the low mobility population of CD47 (111.9 nm, Fig. S7) suggesting that the mobile integrin preferentially associate with the low mobility fraction of CD47 molecules. In support of this, we sometimes observed CD47-αvβ3 co-diffusion events in regions such as microvilli, where the two molecules are co-localized

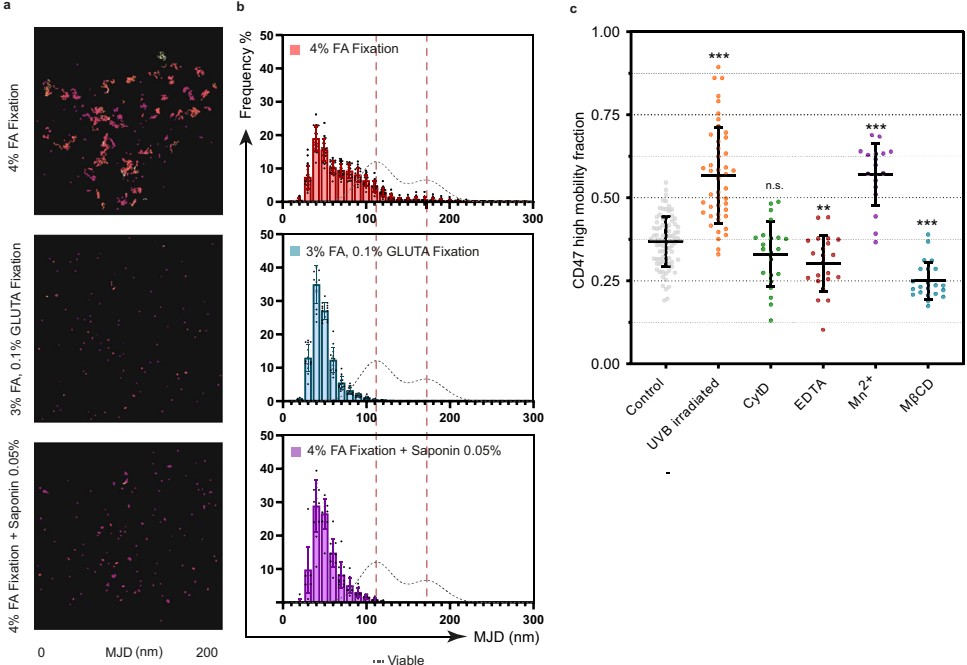

**Fig. 3 Plasma membrane molecules affect the mobility of CD47.** CD47 was detected on viable HeLa cells with an anti-CD47) antibody(B6H12) labeled with A647 and imaged by TIRF microscopy during 60 s (**a,b**) or 120 s (**c**) at 33 Hz. **a** Viable cells were stained to detect CD47 and then fixed: with respectively 4% formaldehyde, 3% formaldehyde and 0.1% glutaraldehyde or 4% formaldehyde followed by 0.05% saponin. Mean Jump Distances (MJDs) was were calculated and each CD47 trajectory was colored according to its MJD value. A range of 250 frames of CD47 tracking was represented. Scale bar, 5 μm. **b** MJD distributions were obtained after merging the MJDs from several cells (4% formaldehyde: $n = 20$ cells, 3% formaldehyde + 0.1% glutaraldehyde: $n = 14$ cells, 4% formaldehyde + 0.05% saponin: $n = 13$ cells) measured in two independent experiments. The dotted red line corresponds to the mean value of the low-mobility (111.9 nm) and high-mobility (172.7 nm) MJDs extracted from viable cell MJD distribution (represented by the gray dotted line). **c** Cells were treated by UVB irradiation ($n = 45$), Cytochalasin D (CytD, $n = 22$), EDTA ($n = 22$), $Mn^{2+}$ ($n = 18$) and MβCD ($n = 22$) and compared to non-treated viable (control) cells ($n = 84$). MJDs were calculated for each cell. Representative individual MJD histograms are shown in Figure S7. The Gaussian distribution of low- and high-mobility fractions was fitted, and the fraction of the high-mobility MJDs relative to the total number of measured MJDs is represented (R-value). Each dot corresponds to one cell and each experiment was performed at least two times independently. The mean and the standard deviation for each treatment are represented. *** denotes $p < 0.0005$, ** denotes $p < 0.005$, and n.s. denotes $p > 0.05$ as determined by a two-sided Student's t test in comparison to the control.

(Supplemental Movie 1, SPT αvβ3/CD47 full frame and Movie#2, SPT αvβ3/CD47 zoomed view).

**Destabilization of cholesterol-containing domains is not sufficient to increase CD47 mobility.** It was previously proposed that cholesterol may be important for CD47/αvβ3 signaling activity[41]. Accordingly, Lv and collaborators showed that CD47 could aggregate in lipid rafts[17], and that destabilization of the latter by methyl-beta-cyclodextrin (MβCD) induced the onset of a diffuse-like pattern of CD47 as observed by immunolabeling. Therefore, we tested the influence of MβCD on the mobility of CD47 by SPT. We observed that this treatment did not increase the fraction of the highly-mobile population of CD47 (Fig. 3c), but that, on the contrary, only the low-mobile fraction remained (Fig. S7). This suggests that the high-mobility state of CD47 is dependent on the presence of cholesterol-rich domains and that the diffuse pattern observed by Lv and collaborators[17] is not associated with an increased mobility of CD47. This conclusion is also strengthened by our above data (Fig. 3a and b) showing that the use of saponin to extract cholesterol drastically diminished the proportion of the still mobile CD47 molecules after fixation.

**Destabilization of cholesterol impacts SIRPα binding and phagocytosis efficiency.** To evaluate how the mobility of CD47 influences its biological function, we first investigated the binding of the SIRPα protein on viable HeLa adherent cells. The binding of a soluble recombinant form of SIRPα (biotinylated-SIRPα/CD172a Fc Chimera) was measured by confocal imaging on HeLa cells treated with $Mn^{2+}$, MβCD or Cytochalasin D (treatment with EDTA could not be performed due to loss of adhesion) (Fig. 5a and b). SIRPα binding was evaluated on individual cells by determining the density (number of spots per μm²) of fluorescence foci. As shown in Fig. 5a and b, the number of SIRPα foci detected per cell drastically decreased after MβCD treatment but not in the other conditions except after incubation with the therapeutic anti-CD47 B6H12 antibody known to block the SIRPα/CD47 interaction. We next tested if the applied treatments had effects on cell engulfment by J774 macrophages (Fig. 5c) using flow cytometry. As expected, the B6H12 anti-CD47 antibody drastically increased phagocytosis. An increase of phagocytosis was also observed after MβCD and to a lesser extent with CytD treatment. The increased phagocytosis after MβCD treatment is in agreement with the observed loss of SIRPα binding (Fig. 5a and b). The apparent increased phagocytosis in the presence of CytD could be biased by the high proportion of cell debris observed with CytD treatment), that can be engulfed independently of the SIRPα/CD47 pathway. The exogenous activation of integrins in HeLa cells by $Mn^{2+}$ did not have a significant effect on phagocytosis, suggesting that activating integrins on the target cell prior to contact with macrophages does not prevent CD47 from acting as a "don't eat-me" signal, in agreement with the observation that SIRPα binding to the cell is preserved upon $Mn^{2+}$ addition (Fig. 5a and b). The increase of

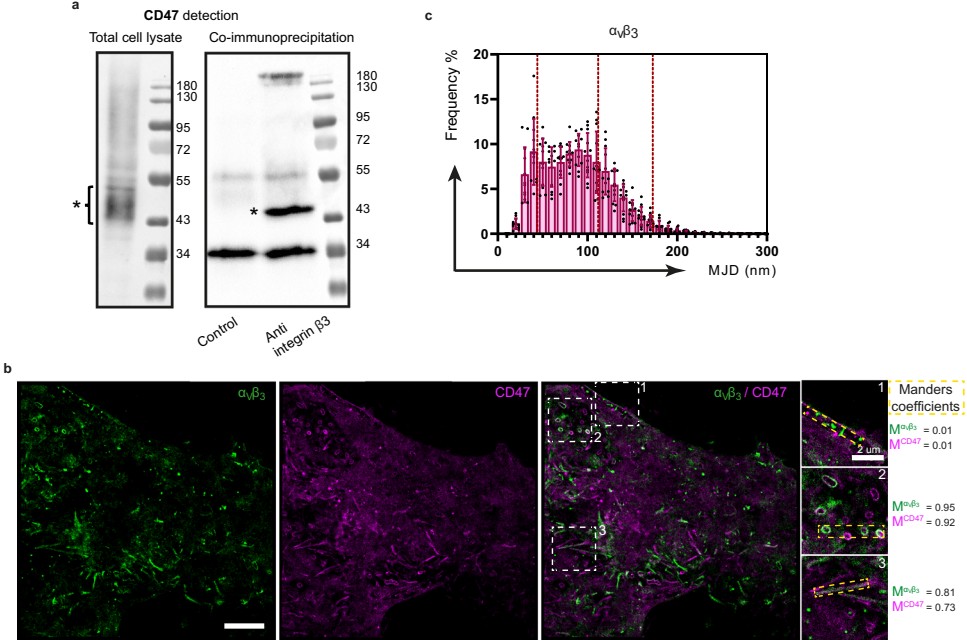

**Fig. 4 Mobile $\alpha_v\beta_3$ integrin and CD47 appear confined in the same membrane area. a** CD47 was detected after β3 integrin subunit immunoprecipitation from a HeLa cell lysate. The cell lysate was absorbed to non-coated magnetic beads (Control) or coated with anti-integrin β3 (AB2984). CD47 was next detected by western blotting with an anti-CD47 antibody (B6H12). As indicated, CD47 detected on a whole cell lysate (total cell lysate) and CD47 detected on the β3 integrin subunit-immunoprecipated sample (co-immunoprecipitation). Samples were submitted to 10 % SDS PAGE analysis under non-reducing conditions. The position of CD47 is indicated by a (*). A non-specific band is detected at 34 kDa in both samples. Molecular weight markers (kDa) are shown. **b** Two color single particle tracking of integrin $\alpha_v\beta_3$ and CD47 was obtained using anti-αVβ3 (23C6) and anti-CD47 (B6H12) antibodies labeled with PE and A647 respectively, and imaged by TIRF microscopy during 120 s at 33 Hz. The localizations of the two proteins were compiled over time to reconstruct super-resolved images. White dotted squares numbered 1, 2, 3 delimit enlarged zones on the right and yellow dotted rectangles delimit zones used to calculate Manders' coefficients. $M^{\alpha v\beta 3}$ and $M^{CD47}$, Manders' overlap coefficient of αvβ3 localization with CD47 and of CD47 with αvβ3, respectively. Scale bar, 5 µm. **c** MJD distribution for αvβ3 extracted from SPT acquisition as previously done in Figs. 2 and 3. Dotted red lines correspond to the CD47 mean MJD value of the immobile population extracted from formaldehyde + glutaraldehyde fixation (44 nm), and from the low mobility (111.9 nm) and high mobility (172.7 nm) population extracted from control viable cells. The MJD distribution was obtained after merging MJDs from several cells ($n = 12$) from two independent experiments.

phagocytosis observed under the various treatments was not due to an effect on cell viability as Annexin V/PI assays didn't show significant apoptosis or necrosis (Fig. S9). All together our observations that (i) removing cholesterol by MβCD increased the phagocytosis efficiency while decreasing the CD47 mobility and SIRPα binding, and (ii) activating integrins did not have an effect on phagocytosis while increasing CD47 mobility, suggest that CD47 mobility and its recognition by the SIRPα receptor are essentially uncoupled.

**Soluble SIRPα does not bind to apoptotic blebs despite the presence of CD47.** Our above results obtained with MβCD treatment on viable cells prompted us to ask whether SIRPα would be able to recognize CD47 differentially according to its location on the apoptotic cell surface. Indeed, apoptosis is accompanied by a decrease of lipid order of the plasma membrane related to phospholipids exchange and cholesterol destabilization[42], inducing profound and heterogeneous structural changes throughout the membrane, a process highlighted by the characteristic membrane blebbing. Interestingly, we observed by confocal microscopy that SIRPα did not bind the characteristic membrane blebs (Fig. 6a and c) despite the presence of CD47 (Fig. 6b and c). This was confirmed at the nanoscale by performing a simultaneous detection of anti CD47 antibody binding and soluble SIRPα binding by STORM imaging. As shown in Fig. 6d, SIRPα binding was detected on apoptotic Hela cells but did not decorate the membrane blebs despite the presence of CD47. Conversely, CD47 and exogenously added SIRPα co-

localized on membrane areas located outside the blebs. This observation suggests that CD47 accessibility to SIRPα is dependent on the way it is exposed on the cell surface and is not directly linked to its mobility.

**Unlike CD47, the mobility of PS facilitates phagocytosis.** In contrast to what we demonstrate for CD47, an increase of PS lateral mobility in the plasma membrane during apoptosis has been suggested to participate in the PS "eat-me" signaling function and was put forward to explain that non-apoptotic cells where PS is constitutively exposed (e.g., B cells or neutrophils)[43] are not engulfed. For this reason, further experiments were done to test the role of PS mobility on the phagocytosis ability of macrophages. We prepared PC/PS phospholipids-coated glass beads as phagocytic targets, using synthetic lipids 1-palmitoyl-2-oleoyl-glycero-3-phosphocholine (POPC) or 1,2-dipalmitoyl-sn-glycero-3-phosphocholine (DPPC) to modulate the 1,2-dioleoyl-sn-glycero-3-phospho-L-serine (DOPS) mobility on reconstituted phospholipids bilayers[44]. TopFluorPS was also incorporated as an indirect marker to monitor the fluidity and the distribution of DOPS lipids. As shown in Fig. 7a and b, the target beads with mobile PS (i.e, containing POPC) as monitored by FRAP analysis (Fig. S10), were markedly more engulfed by J774 macrophages than control beads (i.e, containing DPPC) characterized by a lower PS mobility. This confirms the link between PS mobility and its capacity to serve as an "eat-me" signal toward the receptors expressed by the macrophage.

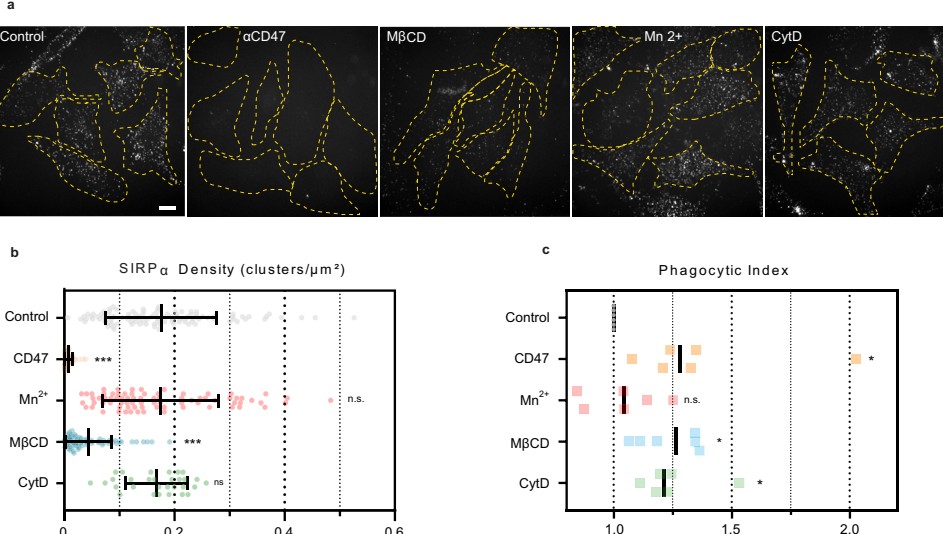

**Fig. 5 SIRPα binding to HeLa cells and their phagocytosis by J774 macrophages are dependent on cholesterol content. a–c** Cholesterol depletion inhibits SIRPα binding and increases phagocytosis. **a, b** Cells were incubated with Cytochalasin D (CytD), Mn2 + , MβCD or anti CD47 antibodies (B6H12). Cells were next incubated with recombinant biotinylated human SIRPα Fc chimera protein, revealed by A647-streptavidin and imaged by confocal microscopy. **a** A Z stack of cells was acquired and the maximum projection is displayed with cells outlined in yellow. Scale bar, 10 µm. **b** Density of SIRPα binding on HeLa treated cells. Each dot corresponds to one cell coming from at least two independent experiments (Control: $n = 106$, anti-CD47: $n = 60$, Mn$^{2+}$: $n = 77$, MβCD: $n = 72$ and CytD: $n = 60$). The mean and the standard deviation for each treatment are represented. *** denotes $p < 0.0005$, and n.s. denotes $p > 0.05$ as determined by a two-sided Student's t test in comparison to the control. **c** Phagocytosis of HeLa treated cells by J774 macrophages. Harvested cells were treated as indicated in the Material and Methods section and washed before adding macrophages. Each point corresponds to an independent experiment ($n = 6$). The median for each treatment is represented.* denotes $p < 0.05$, and n.s. denotes $p > 0.05$ as determined by a two-sided Wilcoxon-Mann-Whitney's test in comparison to the control.

## Discussion

A thorough knowledge of the relative exposure and organization of "eat-me" and "don't eat-me" signals at the plasma membrane is essential to consider therapeutic approaches aimed at the elimination of transformed or apoptotic cells (e.g. in anti-cancer or auto immune disease therapies).

Ecto-CRT has been found to be an "eat me" signal which can also help in the development of an immunogenic response, and CD47 is a crucial "don't eat me" signal, the blocking of which stimulates phagocytosis. There are many remaining questions about how these molecules function together and their potential patterning at the plasma membrane, known to be profoundly reshaped during apoptosis. By accessing the nanoscale thanks to single-molecule localization (SML) microscopy, we analyzed the molecular distribution of CRT, CD47 and PS on viable or apoptotic cells. CRT and CD47 are exposed on viable cells with no obvious co-localization and we only observed their reorganization after apoptosis induction showing a clear redistribution of CRT into clusters at the surface of apoptotic cells with a marked accumulation in blebs. We also noticed that CD47 is present in these blebs. This suggests that modification of CRT/CD47 spatial proximity is neither a prerequisite for activating the calreticulin"eat-me" signal, nor inhibiting the CD47 interaction with the macrophage SIRPα receptor, although this was previously proposed[7]. It was also proposed that CRT and PS could associate in complexes needed for apoptotic cell recognition by phagocytes, based on their reported ability to interact in in vitro assays[19]. However, this was challenged by the fact they are not simultaneously translocated at the cell surface[18,31]. Here, we have shown that PS and CRT are usually not found in the same nanodomains, with a specific segregation of CRT into the blebs but also in peculiar regions outside the blebs where PS is excluded, even if the role of these areas remains to be discovered. Therefore, our observations strongly suggest that when exposed on the outer surface of apoptotic cells, CRT, PS and CD47 can act independently of each other to recognize their phagocytic receptors. Of note, Nilsson and Oldenborg[45] have also analyzed CD47 localization together with PS exposure on apoptotic murine thymocytes by confocal microscopy. Interestingly, they observed that CD47 aggregated in cholesterol-rich GM-1 domains (i.e. possibly RAFT), overlapping with PS or not depending on the apoptotic state of the cell. Even if the underlying mechanism was not clarified, it is however possible that this may have distinct effects on phagocytosis.

Next, we discovered by SPT experiments that CD47 exists in two different mobility states. In healthy adherent HeLa cells, about two-third of the CD47 population is found in a low mobility state, and one-third in a high mobility state. Strikingly this ratio was found to be inverted in early apoptotic cells obtained by UVB irradiation. We wondered if this change toward the high-mobility state could be connected to the decrease of the CD47/SIRPα interaction characteristic of cells going into apoptosis. To provide insight into this question, we investigated the impact of components expected to be altered during apoptosis in affecting the mobility of CD47 and its interaction with SIRPα. Because CD47 is an integrin-associated protein, its activity is presumed to be tuned by the integrin affinity state which is controlled by inside-out or outside-in signaling, a process altered during apoptosis. Accordingly, exogenous destabilization of integrins by EDTA slightly diminished the CD47 mobility, whereas activation of integrins in HeLa cells increased the proportion of highly mobile CD47 at the cell surface. With our two-color SPT experiments, we also showed that some CD47 and αvβ3 integrin molecules were found co-localized in micro-domains putatively assigned to microvilli or vesicles-like domains and could be seen to move together transiently (Fig. 2c and Supplemental Movies 1 and 2). Stationary integrins, in contrast, assumed to be found in focal adhesion

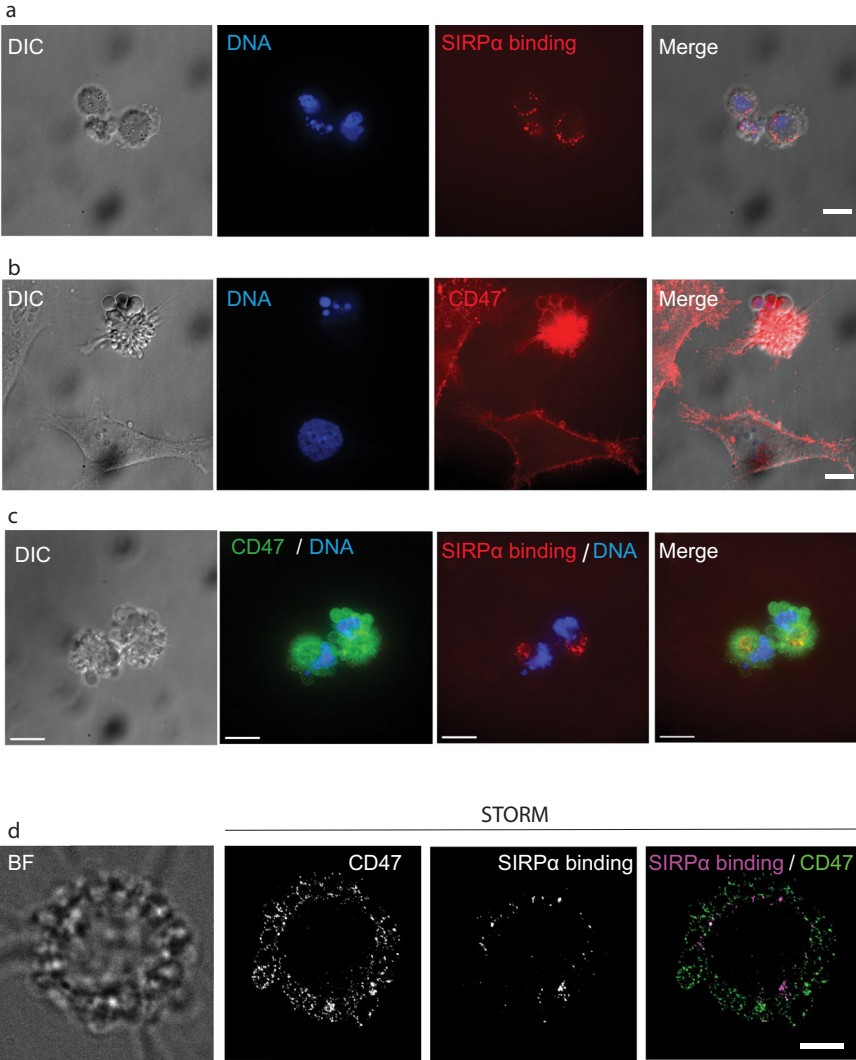

**Fig. 6 Soluble SIRPα does not bind to apoptotic blebs despite the presence of CD47.** Apoptotic HeLa cells were incubated with biotinylated SIRPα and/or anti-CD47 antibody (B6H12). **a** SIRPα binding revealed by A647-streptavidin. **b** Cells were stained with anti-CD47 (B6H12) revealed by secondary antibodies labeled with Cyanine 3 (red). **c** Double labeling by SIRPα and anti-CD47 antibody. Cells were incubated with biotinylated human SIRPα Fc chimera protein, revealed by streptavidin A647 (red) and then fixed and stained with anti-CD47 (B6H12) revealed by secondary antibodies labeled with Alexa 488 (green). In a, b and c samples were mounted with DAPI to detect the DNA (blue) and imaged by spinning disk confocal microscopy. Z stacks of cells were acquired and the maximum projection is displayed. **d** Two-color STORM image showing simultaneous detection of SIRPα binding and CD47 obtained using spectral demixing (see Materials and Methods). SIRPα binding was revealed by A647-streptavidin and anti-CD47 antibody was revealed by a secondary antibody labeled with CF680. Scale bars 10 μm.

sites, did not co-localize with CD47. These observations favor the hypothesis that CD47 associates preferentially with mobile integrin in its low-affinity state. Moreover, our finding that MJDs measured for mobile αvβ3 integrins are similar to those of the low mobile CD47 molecules suggests that the highly mobile CD47 molecules are unlikely to be associated to integrins. Together with our observation that CD47 mobility is enhanced after apoptosis induction, this also suggests that during apoptosis development, CD47 molecules progressively lose their link with integrins. Nevertheless, although integrin activation consequently to $Mn^{2+}$ treatment induced an increase in CD47 mobility, this did not affect SIRPα binding or cell engulfment by macrophages, suggesting that CD47 binding to SIRPα is not directly dependent on integrin regulation by the outside face of the cell. Interestingly it was recently proposed by Morrissey et al[15]. that activating the macrophage integrins eliminates the suppressive CD47 effect through repositioning SIRPα outside of the phagocytic synapse. Conversely, it appears that integrins of

the target cell are not directly involved in tuning the accessibility of CD47 to SIRPα.

Removing cholesterol abolished the highly mobile CD47 population, together with increasing the engulfment of cells by macrophages and decreasing the binding of SIRPα to the cell surface. With the knowledge that depleting cholesterol activates cell death and apoptosis[46,47], this highlights the role of cholesterol to preserve the CD47 "don't eat-me" signal. Cholesterol is involved in the regulation of membrane fluidity and is the major component of lipid rafts, where lipids are more ordered and tightly packed than on the surrounding bilayer. Lv and collaborators[17] have previously suggested that CD47 clustering in cholesterol-rich lipid rafts may be necessary for SIRPα binding. Our data confirm the importance of cholesterol to maintain SIRPα binding but also indicate that CD47 clustering is not critical for SIRPα binding since cells with a large proportion of highly mobile CD47, obtained by activating integrins, are efficiently recognized by SIRPα.

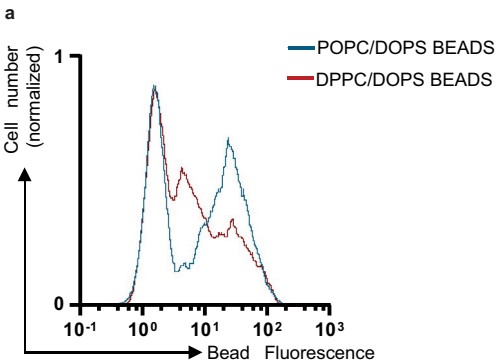

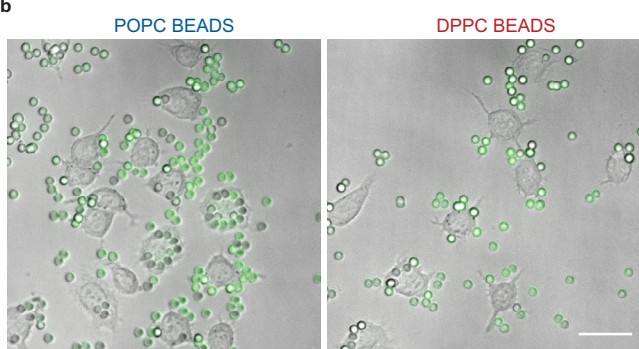

**Fig. 7 PS mobility enhances phagocytosis.** Phagocytosis of lipid bilayer coated silica beads containing DOPS by J774 macrophages. To modulate the mobility of DOPS, the lipid bilayer was composed of POPC (high mobility) or DPPC (low mobility) lipids. **a** J774 were incubated with fluorescent beads (TopFluorPS) and phagocytosis was assessed by measuring the fluorescence of cells with internalized beads by flow cytometry. One representative experiment out of two is shown. **b** Images of beads phagocytosis by J774 macrophages. Scale bar 30 μm.

Apoptosis is marked by changes in cell adhesion resulting from intracellular signaling and cleavage of components of the focal adhesion complex[48,49] together with other major plasma membrane structural alterations. These major modifications can contribute to destabilization of lateral molecular interactions needed for the correct CD47 "don't eat-me" signaling state toward the macrophage. They could induce conformational changes of CD47, altering its binding site to SIRPα or making it inaccessible. Our data showing that SIRPα does not bind to CD47 exposed on apoptotic blebs support the hypothesis that local perturbation of the plasma membrane could induce conformational changes of CD47 or that the SIRPα binding site becomes inaccessible (Fig. 6) potentially due to modification of the lipid environment[41] or to the fluctuation of the plasma membrane flexibility[50]. Of note, such conformational changes were not detected by the used anti-CD47 antibody B6H12, which appeared to bind successfully to the CD47 epitope in all tested conditions. Nevertheless, this hypothesis is in line with data showing the influence of the lipid environment and cholesterol on membrane protein structure and conformation[51,52]. Interestingly, CD47 conformation or epitope modifications were shown at the cell surface of aged erythrocytes promoting TSP-1 recognition and subsequent clearance[27,53].

During the writing of this paper, the whole 3D structure of CD47 was published[54], and two conformational states were observed as a result of the flexibility of the inter-domain connecting the transmembrane domain to the extracellular domain of the molecule. Notably, disturbing a disulfide bond (Cys15-Cys245) increased dynamics of the inter-domain with

concomitant conformational changes in the transmembrane and extracellular domains. Interestingly, destabilization of this disulfide bond impacts CD47-SIRPα interaction[55]. We can thus postulate that, by influencing membrane organization, cholesterol stabilizes the CD47 extracellular domain in one conformation promoting its binding activity.

The opposing effects obtained on the CD47 mobility behavior by destabilizing cholesterol-rich domains or by activating integrins is in agreement with data showing that CD47-αvβ3 interactions leading to integrin clustering are insensitive to cholesterol depletion[56], even if it has been shown that cholesterol is essential for the formation of a multimolecular signaling complex composed of αvβ3, CD47 and G proteins[41]. This underlines the need for future investigations of the CD47/integrin interactions to highlight the role of CD47 in the modulation of integrin affinity or avidity toward its ligands, and associated signaling events.

Overall, the efficacy of the CD47 "don't eat-me" signal does not seem to be directly linked to its molecular mobility behavior in striking contrast to what has been proposed for the PS "eat-me" signal. Indeed it has been suggested that differences in PS mobility could explain why exposure of PS is not sufficient to be an effective "eat-me" signal[43,57] and thus why cells that constitutively express PS are not engulfed[21]. Notably, our results showing that engulfment of lipids-coated beads by macrophages is more efficient when increasing PS mobility, experimentally supports this hypothesis.

To sum up, as schematized in Fig. 8, the present study provides insight into the nanoscale reorganization and dynamics of CD47 induced by apoptosis and will contribute to establish how its cooperative interactions with the macrophage SIRPα receptor is dependent on fluctuations in the composition and properties of the plasma membrane. This may be beneficial to efficiently target the CD47-SIRPα phagocytosis inhibitory signal in cancer immunotherapy research. However, a question that remains to be elucidated is whether the increased diffusion of CD47 on the membrane of apoptotic cells could also be related to a "peculiar reorganization" of cholesterol domains during apoptosis.

## Materials and methods

**Cell culture.** HeLa (CCL2) and J774 (TIB 67) cell lines were provided by the ATCC. The cells were grown in Glutamax DMEM (Thermofisher) supplemented with 10% (v/v) heat-inactivated fetal calfserum, penicillin (0.3 U/ml) and streptomycin (0.3 μg/ml). The cells were tested for Mycoplasma contamination (Mycoalert detection kit, Lonza). All cell lines were maintained at 37 °C, under a 5% CO$_2$ atmosphere. For apoptosis of HeLa cells, cells were grown in sterile dishes overnight to 60–80% confluence and exposed to UVB irradiation (1000 mJ/cm$^2$) at 312 nm in fresh medium (DMEM with fetal calfserum)[58].

**Antibodies and reagents.** The following antibodies and molecules were used (concentrations used are indicated in parentheses): Chicken polyclonal anti-calreticulin (Thermofisher, PA1-902A, 4,4 μg/mL), Rabbit polyclonal anti-ERp57 (Abcam, ab10287, Dilution 1:100), Mouse anti-CD47 conjugated to Alexa Fluor A647 (Santa Cruz, B6H12, 2 μg/mL), Annexin V conjugated to Biotin (Biolegend, 640904, 2.5 μg/mL), Rabbit Polyclonal anti-Chicken IgY labeled with Alexa Fluor 532 (Cohesion Biosciences, CSA3314, 5 μg/mL), Goat polyclonal anti-Chicken labeled with Alexa Fluor 555 (Thermofisher, A21437, 2 μg/mL), Donkey polyclonal anti-Rabbit IgG labeled with FluoProbes 647H (Interchim, FP-SC5110, 10 μg/mL), Streptavidin conjugated to Alexa Fluor 647 (Thermofisher, S32357 - 0,4 μg/mL).

**STORM.** Cells were plated in Nunc™ Lab-Tek™ II Chambered Coverglass (Thermofisher, 155409) at ~50000 cells per well the day prior to fixation. Before seeding, to minimize background fluorescence, the Lab-Tek chambers were cleaned with an ultraviolet Ozone cleaning system (UVOCS®) during 20 min. In addition, to increase the adhesion of apoptotic cells, the Lab-Tek was coated with Poly-D-lysine Hydrobromide (Sigma, P7886) at 0.2 mg/mL during 2 h or overnight at room temperature. All the following steps were performed after fixation except for the phosphatidylserine detection with annexin V-biotin performed in 10 mM Hepes, 150 mM Nacl, 2 mM Ca2+ pH 7.4 during 30 min on ice to avoid internalization. Cells were washed gently with PBS at 37 °C, fixed for 10 min at 37 °C with 3% paraformaldehyde (Electron Microscopy Sciences, 15710) and 0.1% glutaraldehyde (Electron Microscopy Sciences, 16020) in PBS to immobilize the membrane[36].

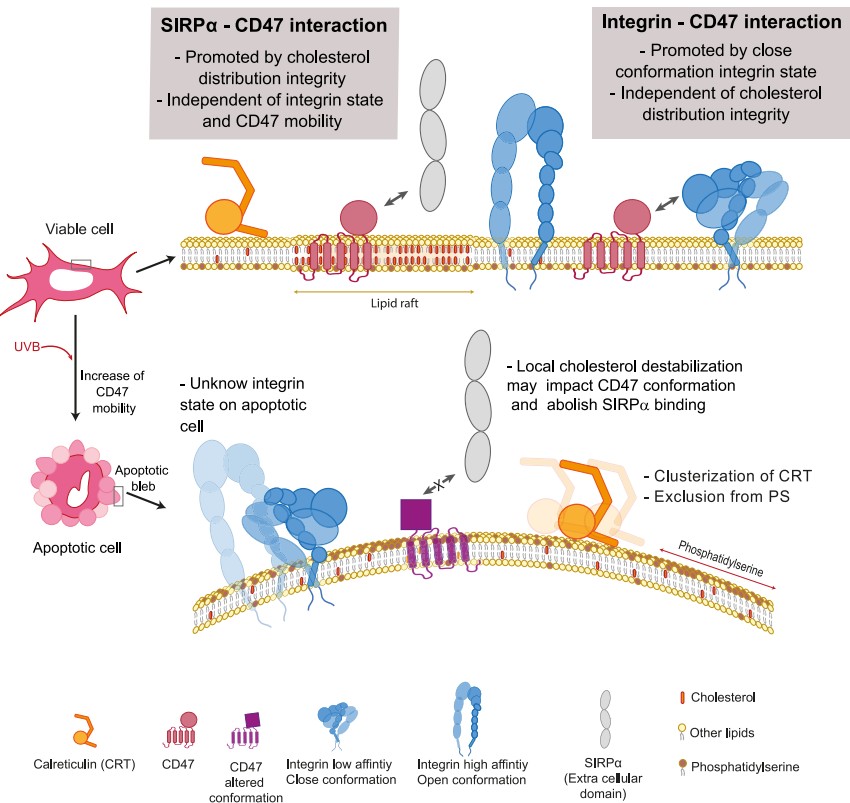

**Fig. 8 Illustration of the modifications on the cell membrane due to apoptosis, involving CD47, PS, CRT, integrins and cholesterol that impact CD47-SIRPα recognition and the exposure of eat-me signals.** On viable cells, the extracellular domain of SIRPα interacts preferentially with CD47 located in high cholesterol content membrane domains. This interaction is independent of CD47 mobility and of the integrin state. When the cell becomes apoptotic, the plasma membrane is altered, leading to phosphatidylserine exposition and destabilization of lipid domains. The cells form apoptotic blebs where calreticulin is relocated and eventually clustered. The destabilization of high cholesterol content domains may potentially promote a conformational change of CD47 and precludes its recognition by the SIRPα receptor. The status of integrin on apoptotic blebs remains to be clarified.

After fixation, cells were washed and incubated for 30 min with 100 μM CuSO4 in ammonium acetate (100 mM, pH 7,4) to react with free aldehyde and to reduce background fluorescence from apoptotic cells[59]. Then, the cells were blocked for 1 hour with 3% bovine serum albumin (Sigma, A7030) in PBS before staining for 1 hour with primary antibodies in PBS, 3% BSA. Thorough washing was performed with PBS, 5 times with 10 min between each washing before incubation with secondary antibodies or streptavidin in PBS, 3% BSA for 45 min at room temperature. Finally, the cells were washed 3 times with PBS and incubated with fiducial markers (sonicated Gold nanoparticles, Sigma, 753688) at 1:8 dilution in PBS, overnight. For Alexa Fluor 555 and FluoProbes 647H secondary antibodies staining, an additional fixation step was performed to stabilize secondary antibodies for 10 min at room temperature with 3% paraformaldehyde and 0.1% glutaraldehyde in PBS. Before imaging, cells were washed gently, immersed with the imaging buffer and the wells were sealed with a coverslip and picodent twinsil® (Picodent, 1300 1000) to limit gas exchange. The imaging buffer was composed of 50 mM Tris, 10% (w/v) D-glucose, 10 mM NaCl, pH 8.0, an oxygen scavenging system composed of 0.5 mg/mL glucose oxidase (Sigma, G2133-50KU), 40 μg/mL catalase (Sigma, C40) and 10 or 100 mM MEA (Cysteamine) according to the dye used. A concentration of 100 mM MEA (Sigma, 30070) was used for Alexa Fluor 532 and 647 and 10 mM was used for Alexa Fluor 555 and FluoProbes 647H.

Cells were imaged on a home-built PALM setup based on an Olympus IX81 inverted microscope equipped with a x100 1.49 NA oil immersion apochromatic objective lens (Olympus). Widefield illumination was achieved by focusing the diode-pumped solid-state 405 nm (CrystaLaser), 532 nm (CNI Laser), 561 nm (Cobolt Jive) and 642-nm laser (Toptica Photonics), beams to the back focal plane of the objective. Laser illuminations intensities were tuned by an acousto-optical tunable filter (AOTF; Quanta Tech). Fluorescence images were acquired onto a 256×256 pixel region of an Evolve 512 back-illuminated EMCCD camera (Photometrics) controlled by the Metamorph software (Molecular Devices). Two-color acquisitions were performed sequentially and 15000 to 20000 frames were recorded for each condition. The following longpass dichroic mirrors were used to reflect the excitation sources listed above: ZT532/640rpc-UF1 (Chroma) for 532 nm and 642 nm, FF580-FDi01-25×36 (semrock) for 561 nm. Fluorescence passed through one of the following bandpass emission filters: ZET532/640 m (Chroma) for A532, A647 and F647H dyes and FF01-630/92-25 (Semrock) for the

A555 dye. The A647 or F647H dyes were imaged first at 20 Hz with a 642 nm laser (2.1 kW x cm⁻² at the beam center). Next the A532 or A555 dyes were imaged at 50 Hz with a 532 nm laser (3.1 kW x cm⁻² at the beam center). For the A555 dye, 561 nm laser (1.8 kW x cm⁻² at the beam center) was used in combination to the 532 nm laser. During data acquisition, a continuous focus system (ZDC, Olympus) was used to maintain a constant focal plane. To correct for chromatic aberration, 100 nm TetraSpeck™ microspheres (Thermofisher, T7279) embedded in the imaging buffer without the scavenging system were imaged in the same conditions as used during acquisition. Image stacks were treated using imageJ. To remove background signal, a temporal median filter was applied on images with a 100 frames interval[60] using the GDSC Single Molecule Light Microscopy (SMLM) ImageJ Plugins (https://github.com/aherbert/gdsc-smlm). After the Gaussian fitting step, we performed drift correction using either gold nanoparticles as fiducial markers, or cross-correlation procedure in a few cases where fiducials could not be identified. We next proceeded with grouping of localizations originating from single molecules detected in subsequent frames to avoid artifactual clustering. To do so we used the following parameters: maximum distance - one-pixel size and trace length -infinite (length of acquisition) as described by Jimenez et al[61]. Additionally, out-of-focus localizations were filtered based on the value of the standard deviation of the Gaussian fit being superior to 200 nm. To allow registration of sequential acquisitions in two-color experiments, the drift displacement of the first channel was applied on the second channel using ChriSTORM (https://github.com/cleterrier/ChriSTORM). The chromatic aberration correction was performed using the Detection of Molecules (DoM) plugin v.1.2.1 for imageJ (https://github.com/ekatrukha/DoM_Utrecht). Finally, estimation of image resolution was performed using Fourier Ring Correlation (FRC) on the whole image using the NanoJ-SQUIRREL imageJ plugin[62]. Voronoï tessellation analysis was computed to quantify cluster size using SR-Tesseler[63,64], and for co-localization analysis Spearman's rank correlation and Manders' coefficients were computed using Coloc-Tesseler[63,64]. Segmentation of the clusters and Manders' coefficients were obtained by applying a threshold based on one time the average density (δ) of the whole dataset. The Spearman's rank analysis was performed on the whole cell and specific areas of 2.6 μm². Control experiments were performed to obtain reference values. An artificially created positive control was obtained by splitting localizations from even and odd frames of an image stack.

A negative control was achieved by vertically flipping the localizations. The experimental positive control was obtained after staining calreticulin with an anti-calreticulin antibody (PA1-902A, 4,4 µg/ml) and two secondary antibodies labeled with dyes of different colors: a rabbit polyclonal antichicken IgY labeled with Alexa Fluor 532 (5 µg/mL) and a goat polyclonal anti-mouse labeled with Alexa Fluor 647 (Thermofisher, A21236, 5 µg/mL) that we found to cross react with IgY from chicken. Two-color STORM imaging of SIRPα binding and CD47 detection was obtained on a SAFe360 Abbelight microscope (M4D cell imaging platform, ISBG) using spectral demixing. Samples were prepared as previously described except that incubation with the recombinant biotinylated SIRPα extracellular domain fusion protein (SIRPα.ex-Fc, R&D, BT4546B-050, 5 µg/mL) was performed for 30 min before paraformaldehyde/glutaraldehyde fixation. To detect biotinylated SIRPα.ex-Fc, cells were washed and incubated with streptavidin Alexa Fluor 647 (Thermofisher, S32357, 5 µg/mL) during 15 min. Then, cells were washed and fixed for 10 min, and the CD47 molecule was labeled with a mouse anti-CD47 antibody (B6H12, 2 µg/mL) and detected with a secondary goat anti-mouse antibody labeled with CF680 (Sigma-Aldrich, SAB4600199, 10 µg/mL).

**Single particle tracking**. Cells were plated in Lab-Tek™ Chambered Coverglass 8-well (Thermofisher, 155411) at ~50000 cells per well the day prior to fixation. Prior to seeding, the Lab-Tek chambers were cleaned with an ultraviolet Ozone cleaning system (UVOCS®) during 20 min. All the following steps were performed at 37 °C under a 5% $CO_2$ atmosphere and in DMEM fluorobrite medium (Thermofisher, A1896701). Cells were washed and then incubated with 1 of the following reagents: 10 mM MβCD (Santa Cruz, sc-215379) during 15 min, 5 µM CytD (Sigma, C8273) during 30 min, or 1 mM Mn2 + (Sigma, M3634) during 30 min. For induction of apoptosis, cells were irradiated directly within the Lab-Tek chambers and stained after 4 h. After treatment, cells were washed and incubated with the anti CD47 (B6H12) antibody labeled with Alexa Fluor 647 at 2 µg/mL during 5 min. Next, cells were washed and imaged in DMEM fluorobrite (containing D-glucose at 4.5 g/L) supplemented with an oxygen scavenging system (0.1 mg/mL glucose oxidase, 8 µg/mL catalase) to reduce oxygen-mediated photobleaching. Finally, to minimize gas exchange, the wells and the lid were sealed with picodent twinsil. For the EDTA condition and to avoid cell detachment during washing steps, cells were imaged directly in PBS with 0.48 mM EDTA (Thermofisher, 15040066) supplemented with D-glucose, glucose oxidase and catalase. For SPT experiments under fixation, cells were washed after primary antibody incubation and next fixed for 10 min at 37 °C with 4% paraformaldehyde (Electron Microscopy Sciences, 15710) in PBS or 3% paraformaldehyde and 0.1% glutaraldehyde (Electron Microscopy Sciences, 16020) in PBS. Cells were then incubated or not with 0.05 % saponin (Sigma, S-4521) in PBS, 3% BSA during 1 h at 37 °C.

Cells were imaged at the coverslip interface using TIRF illumination on the same home-built PALM setup used for STORM microscopy. The temperature was maintained to 37 °C. Depending of the conditions, 2000 to 4000 frames were recorded at 33 Hz with 1% laser of the maximal laser power (give power laser). For two-color single particle tracking acquisitions two Evolve 512 back-illuminated EMCCD camera (Photometrics) were used with a dichroic splitter FF640-FDi02-t3-25*36 (Semrock). Cells were illuminated simultaneously with a 532 nm laser (2.4 W x $cm^{-2}$ at the beam center) and a 642 nm laser (13 W x $cm^{-2}$ at the beam center) and recorded at 33 Hz.

To remove nonuniform fluorescence background without removing the signal originating from non-diffusing molecules, the EVER (Enhanced super-resolution microscopy by extreme value-based emitter recovery) ImageJ plugin was used in a preprocessing step[65,66]. Images were next analyzed with ThunderSTORM[67]. To reconstruct the tracks, SWIFT vsn 0.3.1 was used[68]. Only tracks with more than 10 localizations were kept for further analysis. The MJD (Mean Jump Distance) of each track was calculated and the obtained histograms were fitted with a two-Gaussian distribution model using Prism GraphPad. The amplitude, the mean and the standard deviation of each Gaussian distribution was considered to calculate the area corresponding to each population. Finally, the fraction of the high-mobility population was obtained by dividing the area corresponding to the high-mobility population by that of the total population.

**Confocal microscopy**. Samples were imaged with a laser spinning-disk confocal microscope (Olympus & Andor, M4D cell imaging platform, ISBG). Cells were prepared as for single-particle-tracking experiments, but were fixed afterwards. All steps were performed at 37 °C under a 5% CO2 atmosphere and in DMEM Fluorobrite medium (Thermofisher, A1896701). Cells were first washed gently and then incubated with one of the following reagents: 10 mM MβCD (Santa Cruz, sc-215379) during 15 min, 5 µM CytD (Sigma, C8273) during 30 min, 1 mM Mn2 + (Sigma, M3634) during 15 min, or Mouse anti-CD47 antibody (B6H12, 10 µg/mL) during 30 min. Then, cells were washed and incubated with the recombinant biotinylated SIRPα extracellular domain fusion protein (SIRPα.ex-Fc, R&D, BT4546B-050, 5 µg/mL) during 30 min. To detect biotinylated SIRPα, cells were washed and incubated with streptavidin Alexa Fluor 647 (Thermofisher, S32357, 5 µg/mL) during 15 min. Then, they were washed and fixed for 10 min at 37 °C with 3% paraformaldehyde (Electron Microscopy Sciences, 15710) and 0.1% glutaraldehyde (Electron Microscopy Sciences, 16020) in PBS. Finally, cells were imaged in Fluoromount G with DAPI (SouthernBiotech, 0100-20) using a 60x

objective. To image the whole cells, Z-stacks were collected with a Z step calculated according to the Nyquist criterion. The density of SIRPα labeling was quantified using ImageJ. Briefly, the shape of cells was determined from DIC and auto-fluorescence in 405 nm and 488 nm channels. Maximum projections of Z planes were computed to allow 2D analysis. The "find maxima" function was used to identify each "cluster" according to a fixed threshold. Finally, the density of SIRPα binding was calculated by dividing the number of clusters by the area of the cell. For double labeling of SIRPα and CD47, the SIRPα binding was performed as previously described in the STORM section and the mouse anti-CD47 antibody (B6H12, 2 µg/mL) was detected with a secondary goat anti-mouse antibody labelled with Alexa Fluor 488 (Thermofisher, A-11017, 10 µg/mL). Finally, the cells were imaged in Fluoromount G with DAPI with a 100×1.49 NA objective. Final images were obtained with the visualization software Imaris (Oxford instruments).

For time-lapse imaging of HeLa cell's apoptosis, cells were irradiated directly within the Lab-Tek chambers with UVB and then stained with Hoescht 33342 (Thermofisher, 62249, 5 µg/mL) during 5 min to detect the nucleus. Cells were next imaged in DMEM Fluorobrite supplemented with Annexin V labeled with Alexa Fluor 647 (Biolegend, 640912, 2.5 µg/mL) to detect exposed phosphatidylserine. Cells were visualized 4 h after irradiation with a 60x objective, at 37 °C under a 5% CO2 atmosphere and one image was acquired every 30 s.

**Flow cytometry**. Cells were seeded one day before the collect at 1.5 millions of cells in a P100 dish in complete DMEM. Before UVB irradiation, the medium was changed. Four hours after irradiation, the supernatants of irradiated and non-irradiated cells were collected and adherent cells were harvested with Versene solution (Thermofisher, 15040066). Cells were washed with PBS and then fixed except for the phosphatidylserine detection; in that case, cells were incubated prior fixation with Annexin V biotin in 10 mM Hepes, 150 mM Nacl, 2 mM Ca2+ pH 7.4 during 30 min on ice. Cells were then washed in PBS, fixed for 10 min with 3% paraformaldehyde (Electron Microscopy Sciences, 15710) and 0.1% glutaraldehyde (Electron Microscopy Sciences, 16020) solution at 37 °C. All the following steps were performed at room temperature and separated by washing steps. Cells were blocked with PBS, 3% BSA (Sigma, A7030) for 15 min. Primary antibodies against CRT, ERp57 and CD47 directly labeled with Alexa Fluor 647 in PBS, 3% BSA were added for 45 min at RT. Cells were incubated next in the dark for 30 min with secondary antibodies when necessary and then suspended in PBS for flow cytometry analysis. Controls were done by omitting primary antibodies. Flow cytometry analyses were performed with a MACSQuant VYB cytometer (Miltenyi Biotech, M4D cell imaging platform, ISBG) and at least 10000 events in the analysis gate were collected. Data were treated with MACSQuantify (Miltenyi Biotec) and dot plots were produced with Graphpad Prism 8.

**Phagocytosis assay**. J774 cells were labeled with CFSE (Thermofisher, C34554) and HeLa cells were labeled with PKH26 (Sigma, MINI26-KT) dyes following manufacturer's recommendations. Cells were allowed to adhere overnight. HeLa cells were detached with a Versene solution, washed and incubated with one of the following reagents: 10 mM MβCD (Santa Cruz, sc-215379) during 15 min, 5 µM CytD (Sigma, C8273) during 30 min, 1 mM $Mn^{2+}$ (Sigma, C8273) during 30 min, or mouse anti-CD47 antibody (B6H12) at 10 µg/mL during 30 min. Then, cells were washed and added to J774 macrophages at a ratio of 2:1 (HeLa:J774), for 2 h at 37 °C, 5% CO2 in serum-free DMEM. Cells were then washed and harvested with 0.25% trypsin/EDTA (Thermofisher, 25300054) and kept on ice before analysis by flow cytometry. The percentage (%) of PKH26-HeLa cells in the CFSE-J774 macrophage population was determined for each condition and normalized by the value of phagocytosis of non-treated cells giving the value of the phagocytic index. For each condition, negative controls corresponding to phagocytosis performed at 37 °C in the presence of 5 µM CytD were subtracted. For the anti-CD47 condition, J774 cells were pre-incubated with IgG isotype control at 10 µg/mL (Santa Cruz, sc-3877) to block Fc receptor contribution during phagocytosis.

**Immunoprecipitation and western blot analysis**. Adherent cells were washed twice in cold phosphate-buffered saline and lysed in RIPA buffer (Thermofisher, 89900) (50 mM Tris-HCl pH 8, 150 mM NaCl, 1% Nonidet, P-40, 0.5% sodium deoxycholate and 0.1% sodium dodecyl) with freshly added protease inhibitor mixture (Roche, 5056489001). Cells debris were removed by centrifugation at 9000 g for 20 min at 4 °C. A micro BCA assay was used to quantify protein concentration. Cell lysates were precleared with dynabeads protein G magnetic beads (Thermofisher, 10003D). Immunoprecipitation was performed overnight at 4 °C with 40 µl of dynabeads protein G that were preincubated with 8 µL of a rabbit anti-β3 integrin subunit antibody (Sigma, AB2984, 1 mg/ml). After washing the beads with the lysis buffer, immunoprecipitated proteins were eluted with a SDS sample buffer under non-reducing conditions. They were analyzed by western blotting after separation on 10% SDS-polyacrylamide gel electrophoresis using the mouse monoclonal anti-CD47 antibody (B6H12, Santa Cruz, sc-12730, dilution 1:200) and a horseradish peroxidase-conjugated secondary antibody for chemiluminescence detection (Sigma, A0545, dilution 1:3000).

**Lipid-coated beads and phagocytosis**. Silica 5 µm beads (Bang laboratories, ssd5003) and borosilicate glass vials were cleaned with an ultraviolet Ozone

cleaning system (UVOCS®) during 10 min. Lipids in chloroform were transferred into the glass vial at the following proportions: 89% POPC (Avanti polar lipids, 850457 C) or DPPC (Avanti polar lipids, 850355 C), 10% DOPS (Avanti polar lipids, 840035 C) and 1% TopFluorPS (Fluorescent PS analog to allow fluorescence detection, Avanti polar lipids, 810283 C) and dried under nitrogen gas. Then they were suspended in sterile DPBS at 2 mg/mL, quickly vortexed and sonicated during 20 min. Then silica beads in PBS were added to the lipids in order to obtain a concentration of 1 mg/mL lipids and 1% w/vol beads. The mix was sonicated during 15 min and next left under agitation during 30 min. Finally, the beads were washed two times and suspended in PBS to 1% w/vol beads. Uniform lipid coating was checked by flow cytometry. Beads were added to J774 macrophages at a ratio of 10:1 (beads:J774), for 1 h at 37 °C, 5% CO2 in serum-free DMEM. J774 cells were then washed and harvested with 0.25% trypsin/EDTA and kept on ice before analysis by flow cytometry. To assess the global lipid mobility reduction within DPPC in comparison to POPC bilayers, we monitored the mobility of the fluorescent PS analog TopFluorPS by FRAP experiments performed with a spinning disk confocal microscope (Olympus & Andor, M4D cell imaging platform, ISBG). Data were analyzed with EasyFRAP-web[69].

**Statistics and reproducibility**. The details of each quantification during super-resolution image processing are included in the Methods section. Statistical analysis was performed in Prism 8 (GraphPad). The results are expressed as mean ± standard deviation (SD).The statistical tests used, the sample sizes and number of independent experiments are mentioned in the figure legends. Two sided $P$ values <0.05 were considered statistically significant.

**Figure illustration**. The Fig. 8 was partly generated using Servier Medical Art, provided by Servier, licensed under a Creative Commons Attribution 3.0 unported license (https://creativecommons.org/licenses/by/3.0/).

**Reporting summary**. Further information on research design is available in the Nature Portfolio Reporting Summary linked to this article.

## Data availability
Complete imaging datasets are available from the Lead Contact. Data used in the figures are available on figshare https://doi.org/10.6084/m9.figshare.21915318.

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

## Acknowledgements

This work used M4D facilities at the Grenoble Instruct-ERIC Center (ISBG; UMS 3518 CNRS CEA-UGA-EMBL) with support from the French Infrastructure for Integrated Structural Biology (FRISBI; ANR-10-INSB-05-02) and GRAL, a project of the University Grenoble Alpes graduate school (Ecoles Universitaires de Recherche) CBH-EUR-GS (ANR-17-EURE-0003) within the Grenoble Partnership for Structural Biology. Research reported in this manuscript was supported by the French National Research Agency (grant ANR-16-CE11-0019) and GRAL (7C047GRAL). IBS acknowledges integration into the Interdisciplinary Research Institute of Grenoble (IRIG, CEA. We are grateful to Dr Ulrike Endesfelder (Max Planck Institute for Terrestrial Microbiology, Marburg, Germany) to provide us the SWIFT tracking software. We thank Dr. Joanna Timmins for critical reading of the manuscript.

## Author contributions

Conceptualization, S.D., D.B. and P.F.; Methodology, S.D., P.F, P. T-D. and D.B.; Investigation, S.D., P.T-D., O. G. and J-P. K.; Validation, P.F. and D.B. Writing – Original Draft Preparation P.F and S.D; Visualization, S.D. and P.F.; Funding Acquisition, N.T. and P.F.; Writing – Review & Editing, P.F., D.B, J-P. K., N.T. and D.B.; Supervision, P.F. and D.B.

## Competing interests

The authors declare no competing interests.
