## [Peer Review File · Communications Biology]

Reviewers' comments:

Reviewer #1 (Remarks to the Author):

Summary: CD47 is a 'Don't eat me' signal present on viable cells and overexpressed on cancer cells. Previous studies (Lv et al) have shown that CD47 is still present on apoptotic cells at the time of engulfment. The question of how CD47 is inactivated to allow for engulfment of apoptotic cells remains an exciting unanswered question, that the authors of this paper attempt to address. Previous work has suggested that CD47 is clustered on healthy cells and becomes diffuse on apoptotic cells (Lv et al). The authors of this previous study suggested that the higher avidity interaction between SIRPA and clustered CD47 would more potently prevent phagocytosis. Another study (Nilsson et al DOI: 10.1016/j.bbrc.2009.06.121) has examined CD47 and PS localization specifically but their results are somewhat contradictory to the Lv et al study.

The strength of this paper is examining CD47 mobility and how mobility is affected by apoptosis, cholesterol and integrin activation. The weakness of this paper is that many conclusions are supported by only a single piece of data (detailed in major points). Multiple experiments that build on each other would strengthen my confidence that these findings are robust and valid. Further, I am largely not sure how these mobility changes affect phagocytosis.

Major points:

I am confused about the conclusions being drawn from Figure 1 and S2. These figures contain data from a single cell per condition, which seems insufficient. Is there heterogeneity cell to cell? Or in different stages of apoptosis? How were the cells selected to remove bias? How were the subcellular regions selected for analysis? I may be missing the point the authors are trying to make with this data, as it seems inconclusive to me.

Figure 3a-b is difficult to interpret because fixing with these chemicals changes so much about the cell and its membrane. The altered mobility could be because some of the CD47 is fixed to adjacent molecules, or because the fixed proteins serve as barriers that limit protein mobility, or because the lipid composition is altered. In addition, the mobility is dramatically different than in unfixed cells, and the shifts in mobility are relatively subtle compared to the unfixed condition. It raises the questions of how meaningful these shifts are. I would feel more confident if additional control proteins were analyzed. Some indication of cell-to-cell variability or statistical analysis may be helpful as well.

Figure 4B also requires some explanation of how subcellular regions are being selected for co-localization analysis. If the authors are concluding that CD47 and integrin interact in specific subcellular regions, that is interesting but requires more in depth analysis across multiple cells. Examining co-localization with markers for various subcellular compartments could be an interesting way to add to this paper. For example, it would be simple to also stain for a marker for focal adhesions to confirm the authors suggestion that these regions lack CD47.

The phagocytosis assays in Figure 5b require additional controls. As the data currently stands, its quite possible that these treatments enhance phagocytosis by simply making the cells sicker, rather than by altering CD47 mobility. For example, is PS disrupted? Does removing CD47 from hela cells eliminate these effects? A knockout of CD47 in hela cells should be a simple way to verify this.

The authors propose that cholesterol is required for SIRPA binding. Why would cholesterol/lipid rafts affect SIRPA monomer binding? Is the affect a direct affect of cholesterol, or an indirect effect of a sicker cell? The previous hypothesis was that CD47 clustering provided high avidity binding between cells with multiple SIRPA receptors. However this does not explain the observed difference in monomer binding.

The authors suggest that CD47 mobility does not affect SIRPA binding based on data analyzing SIRPA

monomer binding. In a cellular context, where SIRPA is in a mobile plasma membrane and may form higher avidity interactions with CD47 as suggested by Lv et al, I am not sure this result would be the same. If mobility does not affect SIRPA binding, does it affect phagocytosis at all and if so how? In their discussion the authors postulate mobility does not affect phagocytosis (if I am interpreting lines 392-398 correctly) so I am not certain how the observed changes in CD47 mobility fit into a larger biological context.

Minor points

Does binding of the PS probes (annexin for example) alter the nanoscale localization of PS?

In addition to the extremely informative Lv et al paper, CD47 localization during apoptosis was also examined in Nilsson and Oldenborg <https://doi.org/10.1016/j.bbrc.2009.06.121>

Reviewer #2 (Remarks to the Author):

The study by Samy Dufour et al. demonstrates how plasma membrane modifications of CD47 shape apoptotic 2 cell recognition. By deploying STORM imaging and single-particle tracking, the authors achieved nanoscale imaging analysis of CD47 and its role in mediating CD47-SIRPa interaction, cell apoptosis and phagocytosis of apoptotic cells. In general, the experiment design is logic and data are solid. Although largely in line with previous studies, the results derived from this manuscript provide certain novel information for our understanding CD47-SIRPa interaction and its dynamic modulation during cell apoptosis and phagocytosis. Two issues are needed to be addressed.

- 1) The authors claimed "Soluble SIRP α does not bind to apoptotic blebs", however, the data supporting this conclusion is weak. Given that this statement is important for CD47-SIRPa interaction field, more evidence should be provided.
- 2) In general, the quantitative analysis is relatively weak. The authors may refine the quantitative method or examine more image to narrow down the error bar.

Reviewer #3 (Remarks to the Author):

The authors study how the mobility of CD47, an important "don't eat me" signal, and its localization relative to related signaling molecule correlate and potentially affect the recognition of apoptotic cells by macrophages. For that, they combine (direct) STORM imaging with SPT. They also use an extensive range of perturbations to the PM.

Focusing on technical aspects of the imaging and analyses - the SPT imaging seems well-developed and properly analyzed. Still, I have major concerns regarding the very low number of STORM-acquired cell images, and their statistical analyses (see details below). The related issues compromise the detailed conclusions of the paper.

General -

* It seems that all STORM related data are restricted to 1 (or 2) cells (I couldn't find specific cell

counts for these experiments). Rather, the analyses are performed on multiple regions. The authors should show a much larger cell count for interpreting the STORM images (as they do in the SPT experiments). This is essential for accounting for cell-cell variability.

* How did the the authors group their localizations (over space and time) in STORM imaging sequences?

This step is needed to control for multiple artifacts associated with single molecule localization microscopy in general, and esp. for STORM; incl. fluorophore blinking and presence of multiple fluorophores on individual antibody molecules. For that, detailed understanding of fluorophore blinking statistics is needed. The lack of details suggest that the authors did not go through this critical analysis step.

Also, target molecules could be highlighted by more than one antibody - esp. when using primary and secondary antibodies for labeling. How could this affect the results?

* The authors use Spearman's (or sometimes Mander's) statistics for colocalization. These analyses are common for analyses of diffraction limited microscopy images. However, there are well established statistics for single molecule localization microscopy, e.g. coordinate based colocalization (CBC) or bivariate pair-correlation functions (PCFs) or Ripley's functions. The authors should show how their results look like when using these statistics.

* Apoptotic cells tend to be fragmented and scatter broadly the excitation light. The treatment of free aldehydes is appreciated.

Still, the authors should show that the apoptotic cells do not show such scattering in unlabeled cells, as compared to labeled cells. This control is needed in each of the spectral regions used for fluorescence detection.

Fig. 1 - The authors should also use non-specific markers (not just ERp57) as negative controls. A non-specific marker of membrane could be useful.

Fig. 4 - What is the origin of the specific band at 180kDa? How are the non-specific bands at 34kDa explained?

Fig. 4 - The identification of restricted areas as microvilli or vesicle domains is not convincing. These domains could be caveolae or other PM-related structures. The authors are advised to use non-specific PM markers or other more specific markers (actin) to understand the nature of these domains. Can the authors employ 3D STORM to better visualize these structures?

Fig. 6 - I suggest that the authors specify how the different interactions depend on the perturbations? (i.e. instead of description via 'dependent', specify if the interaction is actually elevated or diminished)

Point by point responses to Reviewers' comments: Ref :COMMSBIO-21-3491A

We thank the reviewers for the initial reading and comments on our manuscript.

Reviewer #1 (Remarks to the Author):

Summary: CD47 is a 'Don't eat me' signal present on viable cells and overexpressed on cancer cells. Previous studies (Lv et al) have shown that CD47 is still present on apoptotic cells at the time of engulfment. The question of how CD47 is inactivated to allow for engulfment of apoptotic cells remains an exciting unanswered question, that the authors of this paper attempt to address. Previous work has suggested that CD47 is clustered on healthy cells and becomes diffuse on apoptotic cells (Lv et al). The authors of this previous study suggested that the higher avidity interaction between SIRPA and clustered CD47 would more potently prevent phagocytosis. Another study (Nilsson et al DOI:10.1016/j.bbrc.2009.06.121) has examined CD47 and PS localization specifically but their results are somewhat contradictory to the Lv et al study.

The strength of this paper is examining CD47 mobility and how mobility is affected by apoptosis, cholesterol and integrin activation. The weakness of this paper is that many conclusions are supported by only a single piece of data (detailed in major points). Multiple experiments that build on each other would strengthen my confidence that these findings are robust and valid. Further, I am largely not sure how these mobility changes affect phagocytosis.

Major points:

I am confused about the conclusions being drawn from Figure 1 and S2. These figures contain data from a single cell per condition, which seems insufficient.

Is there heterogeneity cell to cell? Or in different stages of apoptosis?

How were the cells selected to remove bias?

How were the subcellular regions selected for analysis? I may be missing the point the authors are trying to make with this data, as it seems inconclusive to me.

At first, this analysis was designed to compare cells at different stages of apoptosis. As noted by the reviewer, on Fig1 and S2 of the submitted manuscript single cells have been selected, and co-localization statistics (panel e) have been derived from several ROIs within individual cells. These selected cells were representative of apoptotic phenotypes based on the presence of typical membrane shrinkage and blebbing associated to apoptosis as described in Figure S1. The limited number of cells presented was also due to the low proportion of cells showing an apoptotic phenotype at the early stage of apoptosis. ROI were selected where the two labels were detected and thus where co-localization could be measured. Moreover, for each cell the co-localization was measured on the whole cell surface to evaluate the global evolution of the localization and is thus independent of the manner we have chosen the ROI.

However, we agree that data from a larger number of cells are necessary to reinforce our conclusions. In consequence, we added localization data obtained from new images acquired in the same conditions. The localization statistics are now presented for three cells for each condition. All additional images and their details are shown in Supplemental figure 4.

Overall, the message we would like to convey with this data is: co localization of CRT with PS and CD47 is weak compared to that measured with ERp57, thus underlining in particular that the modulation of the co localization of CRT with CD47 might not be a crucial point in the early stages of apoptosis, contrary to what has been suggested previously by Gardai et al (*Cell* **123**, 321–334 (2005)). Accordingly, we modified the text lines 149-153.

Figure 3a-b is difficult to interpret because fixing with these chemicals changes so much about the cell and its membrane. The altered mobility could be because some of the CD47 is fixed to adjacent molecules, or because the fixed proteins serve as barriers that limit protein mobility, or because the lipid composition is altered.

Indeed, we agree that chemical fixation alters a lot the cell and its membrane. However, this experiment was designed to identify some clues about CD47 mobility as observed on viable cells. We first noticed the immobilization of only one population of CD47 with formaldehyde (FA) treatment, confirming the heterogeneity of CD47 population. Then, in combination with formaldehyde, we added glutaraldehyde known to strongly immobilize lipids, or saponin to extract cholesterol. In the presence of saponin or glutaraldehyde, the CD47 mobility is drastically diminished, suggesting that the residual mobility of CD47 observed on FA fixed cells is linked to a lipid component of the membrane: the cholesterol.

In addition, the mobility is dramatically different than in unfixed cells, and the shifts in mobility are relatively subtle compared to the unfixed condition. It raises the questions of how meaningful these shifts are. I would feel more confident if additional control proteins were analyzed.

From the literature, it is known that membrane-associated macromolecules are still mobile even after chemical fixation with formaldehyde (Tanaka et al, 2010, <https://doi.org/10.1038/nmeth.f.314>). In the case of CD47, whereas mobility is strongly reduced (but not suppressed) on FA-fixed samples compared to non-fixed ones, addition of saponin or glutaraldehyde clearly makes the residual low mobility population disappear. For a better presentation of the data, we have now added on the 3 graphs the “fit” obtained with the viable cells (figure 3b). We think that these experiments are in line with a role of cholesterol in controlling CD47 mobility.

Importantly, changes in diffusive behavior after fixation is dependent on the type of membrane protein tracked. This makes the use of a control protein, as suggested by the reviewer, difficult.

Some indication of cell-to-cell variability or statistical analysis may be helpful as well.

The histogram shown in figure 3b is based on the merging of a quite large number of cells ($n > 10$ in all cases) as described in the legend. In consequence, the relative error on each bar of the histogram properly accounts for the variability between cells.

Figure 4B also requires some explanations of how subcellular regions are being selected for co-localization analysis. If the authors are concluding that CD47 and integrin interact in specific subcellular regions, that is interesting but requires more in depth analysis across multiple cells. Examining co-localization with markers for various subcellular compartments could be an interesting way to add to this paper. For example, it would be simple to also stain for a marker for focal adhesions to confirm the authors suggestion that these regions lack CD47.

As suggested, we have performed immunofluorescence labelling for CD47 or integrin $\alpha v \beta 3$ and the focal adhesion protein vinculin. CD47 was not enriched in focal contact area when cells are spread on vitronectin while integrin is detected and strongly colocalized. We added this information in the text (lines 219-221) and added corresponding confocal images in supplementary figure S7.

The phagocytosis assays in Figure 5b require additional controls. As the data currently stands, its quite possible that these treatments enhance phagocytosis by simply making the cells sicker, rather than by altering CD47 mobility. For example, is PS disrupted?

We agree with the reviewer that the treatments could have induced cell apoptosis. We had checked this by an Annexin V/PI analysis by flow cytometry, on cell samples used in our phagocytosis assay. These data are now added in supplementary figure S8 and we added the information in the manuscript (lines 268-270). The treatments do not have any noticeable effect on cells death except for Mn^{2+} for which we detected a slight increase without significant change in the phagocytosis (fig 5c).

Does removing CD47 from hela cells eliminate these effects? A knockout of CD47 in hela cells should be a simple way to verify this.

Indeed, a KO of CD47 is another way to study this, but it would have required the use of genetically engineered cells or siRNA transfection, which could have led to further bias and the need for additional controls. We believe that the use of anti-CD47 antibodies, well known to block the CD47 function, is an appropriate control allowing the use of unmodified cells.

The authors propose that cholesterol is required for SIRPA binding. Why would cholesterol/lipid rafts affect SIRPA monomer binding? Is the affect a direct affect of cholesterol, or an indirect effect of a sicker cell? The previous hypothesis was that CD47 clustering provided high avidity binding between cells with multiple SIRPA receptors. However this does not explain the observed difference in monomer binding. The authors suggest that CD47 mobility does not affect SIRPA binding based on data analyzing SIRPA monomer binding. In a cellular context, where SIRPA is in a mobile plasma membrane and may form higher avidity interactions with CD47 as suggested by *Lv et al.*, I am not sure this result would be the same. If mobility does not affect SIRPA binding, does it affect phagocytosis at all and if so how? In their discussion the authors postulate mobility does not affect phagocytosis (if I am interpreting lines 392-398 correctly) so I am not certain how the observed changes in CD47 mobility fit into a larger biological context.

We thank the reviewer for this comment. First, we would like to stress out that the recombinant human SIRP α /CD172a used in this study is a disulfide-linked homodimer.

Lv et al., hypothesized that the clustering of CD47 in large domains at the cell surface promotes high avidity binding of SIRP α . During apoptosis this distribution of CD47 could be destabilized, therefore decreasing the binding of SIRP α .

However, our STORM observations (Fig 1a) did not reveal such organization of CD47 at the membrane of HeLa cells. Instead, we observed punctuated and homogenous distribution of CD47 at the cell surface on both viable and apoptotic cells. In order to complement these observations, we studied the mobility of CD47 during apoptosis. Even if we observed an

increased mobility of CD47 at the membrane of apoptotic cell, our results demonstrate no dependence between the mobility of CD47 and its recognition by SIRP α .

Therefore, based on our observations (i) that loss of cholesterol decreased SIRP α binding to the cell, (ii) that these cells are efficiently engulfed and (iii) that the apoptotic blebs are not recognized by SIRP α despite the presence of CD47, we conclude that changes in the local environment of CD47 modulate its capacity to impair phagocytosis through altered interaction with SIRP α . We postulate that cholesterol might indirectly affect SIRP α binding through a putative conformational change of CD47, as detailed in the discussion section of the manuscript.

We thus speculate that the increased mobility of CD47 in apoptotic blebs arises from a general increase of cell membrane mobility during apoptosis. In contrast to CD47, this membrane property modification could promote PS recognition on the cell membrane as shown on Figure 7 using a bead model. Plasma membrane alteration would therefore promote apoptotic cell recognition and clearance by decreasing the capacity of CD47 to interact with its partner SIRP α and increasing the recognition of PS by its associated receptors.

Minor points

Does binding of the PS probes (annexin for example) alter the nanoscale localization of PS?

It is indeed not excluded that annexin V may influence the nanoscale localization of PS as could be the case for other labeling strategies on unfixed cells. For this reason, we stained apoptotic cells on ice to minimize annexin V internalization and membrane restructuring.

In addition to the extremely informative Lv et al paper, CD47 localization during apoptosis was also examined in Nilsson and Oldenborg <https://doi.org/10.1016/j.bbrc.2009.06.121>

We thank the reviewer for pointing this out; the paper of Nilsson and Oldenborg is now cited and their results, in particular the partial co-localization between CD47 and PS, is commented (discussion section, lines 332-337).

Additional information: Due to addition of new images in the initial Figure 5, we have split its content into three figures (Figures 5-7):

The new Figure 5, corresponding to the panels a, b and c of the previous Figure 5.

Figure 6, containing the images.

Figure 7 contains the data previously presented in Figure 5 (e and f).

Figure 6 contains the images.

Figure 8 corresponds to the previous Figure 6

Reviewer #2 (Remarks to the Author):

The study by Samy Dufour et al. demonstrates how plasma membrane modifications of CD47 shape apoptotic cell recognition. By deploying STORM imaging and single-particle tracking, the authors achieved nanoscale imaging analysis of CD47 and its role in mediating CD47-SIRP α interaction, cell apoptosis and phagocytosis of apoptotic cells. In general, the experiment design is logic and data are solid. Although largely in line with previous studies, the results

derived from this manuscript provide certain novel information for our understanding CD47-SIRP α interaction and its dynamic modulation during cell apoptosis and phagocytosis. Two issues are needed to be addressed.

1) The authors claimed “Soluble SIRP α does not bind to apoptotic blebs”, however, the data supporting this conclusion is weak. Given that this statement is important for CD47-SIRP α interaction field, more evidence should be provided.

The previous confocal images are now completed by additional experiments shown in a new figure (Figure 6). We have added one-color 2D STORM images of SIRP α binding and CD47 distribution on apoptotic HeLa cells (Panels a and b) and we have also obtained a two-color SIRP α /CD47 STORM image that shows no significant co localization. We believe that this will convince that SIRP α does not bind efficiently to the apoptotic blebs despite the presence of CD47.

Additional information: Due to addition of new images in the initial Figure 5, we have split its content into three figures (Figures 5-7):

The new Figure 5, corresponding to the panels a, b and c of the previous Figure 5.

Figure 7 contains the data previously presented in Figure 5 (e and f).

Figure 6 contains the images.

Figure 8 corresponds to the previous Figure 6

2) In general, the quantitative analysis is relatively weak. The authors may refine the quantitative method or examine more image to narrow down the error bar.

For STORM super resolution images on viable or apoptotic cells (Figure 1), we agree that a larger number of cells was needed to reinforce our conclusions; we thus added new data obtained from several cells and acquired for each condition. Localization statistics are now derived from the analysis of three cells for each condition. Co localization statistics (panel e) are from several ROI ($n \geq 7$) per cell.

For CD47 mobility analysis, all experiments were reproduced for quite many cells in each condition, so that we believe that the provided statistics are sufficient to draw the conclusions discussed. As specified in the legend of figure 3, for CD47 MJD analysis, 13 to 20 cells were collected (panel b). In panel c each spot corresponds to the data obtained from single cells measured from two independent experiments. For MJD analysis of integrin $\alpha v \beta 3$ mobility (figure 4) $n = 12$. For SIRP α binding (Fig 5) 60 to 104 cells were analyzed and for phagocytosis assays 6 independent experiments were done. All data will be made accessible to the reader.

Reviewer #3 (Remarks to the Author):

The authors study how the mobility of CD47, an important "don't eat me" signal, and its localization relative to related signaling molecule correlate and potentially affect the recognition of apoptotic cells by macrophages. For that, they combine (direct) STORM imaging with SPT. They also use an extensive range of perturbations to the PM. Focusing on technical aspects of the imaging and analyses - the SPT imaging seems well-developed and properly analyzed. Still, I have major concerns regarding the very low number

of STORM-acquired cell images, and their statistical analyses (see details below). The related issues compromise the detailed conclusions of the paper.

General

-

* It seems that all STORM related data are restricted to 1 (or 2) cells (I couldn't find specific cell counts for these experiments). Rather, the analyses are performed on multiple regions. The authors should show a much larger cell count for interpreting the STORM images (as they do in the SPT experiments). This is essential for accounting for cell-cell variability.

We have increased the number of cells analyzed in figure 1. Localization statistics are now shown for the analysis of three cells for each condition. Co-localization statistics (panel e) are from several ROI ($n \geq 7$) of 3 cells by conditions.

* How did the the authors group their localizations (over space and time) in STORM imaging sequences? This step is needed to control for multiple artifacts associated with single molecule localization microscopy in general, and esp. for STORM; incl. fluorophore blinking and presence of multiple fluorophores on individual antibody molecules. For that, detailed understanding of fluorophore blinking statistics is needed. The lack of details suggest that the authors did not go through this critical analysis step.

We thank the reviewer for pointing this out. We are aware of the complex blinking behavior of the employed fluorophores, as well as the fact that multiple fluorophores may bind single antibodies (degree of labeling > 1). The procedure we used was standard, following that described by Jimenez et al in a recently published paper (Methods 2020,174, 100-114., <https://doi.org/10.1016/j.ymeth.2019.05.008>). Repeated localizations of single fluorophores certainly remain present in our data, as in most STORM-based images. Due to this, we refrained attempting any quantitative molecular counting in the present work, and our conclusions are not based on any such analysis. Importantly, all localizations data were analyzed under identical acquisition and processing strategies so that they can be safely compared. As suggested by the reviewer we have added information to the Material and Method section (line 789-796).

“After the Gaussian fitting step, we performed drift correction using either gold nanoparticles as fiducial markers, or cross correlation procedure in a few cases where fiducials could not be identified. We next proceeded with grouping of localizations originating from single molecules detected in subsequent frames to avoid artifactual clustering. To do so we used the following parameters: maximum distance - one pixel size and trace length -infinite (length of acquisition) as described by Jimenez et al ⁶¹. Additionally, out-of-focus localizations were filtered based on the value of the standard deviation of the Gaussian fit being superior to 200 nm”.

Of note, when multiple fluorophores bind to a single antibody, they tend to collectively behave as a single fluorophore, due to complex energy transfer phenomena as recently described (ACS Nano 2020, 14, 10, 12629–12641 <https://dx.doi.org/10.1021/acsnano.0c06099>), so that this is not expected to greatly influence the blinking behavior of STORM data.

Following the reviewer’s remark, we wondered how postprocessing of our data could influence the outcome of our super resolution images. To this aim, we modified both the search distance

(range 20-120 nm) and the number of off-frames (range 0-2) used for grouping localizations, and found no significant difference.

Also, target molecules could be highlighted by more than one antibody - esp. when using primary and secondary antibodies for labeling. How could this affect the results?

We are aware of these potential problems but using primary and secondary antibodies remains a common approach in the super-resolution field, despite well-known stoichiometry issues and linkage errors. Often, the choice is limited. Nevertheless, we were sometimes able to test several combinations. For detecting calreticulin, anti-chicken AF 532 or anti-chicken AF 555 (in the case of double CRT/ERP57 labeling, see supplementary data) were used with similar results. To detect CD47 we have obtained similar results with two primary monoclonal antibodies (B6H12 and CC2C6) and using an AF 647 secondary antibody. Similar results were also obtained more recently with the two anti-CD47 mAbs and using a secondary anti mouse IgG antibody coupled to CF680 to replace AF 647-coupled antibodies.

* The authors use Spearman's (or sometimes Mander's) statistics for colocalization. These analyses are common for analyses of diffraction limited microscopy images. However, there are well established statistics for single molecule localization microscopy, e.g. coordinate based colocalization (CBC) or bivariate pair-correlation functions (PCFs) or Ripley's functions. The authors should show how their results look like when using these statistics.

We agree with the reviewer that multicolor SMLM co-localization quantification requires specific procedures due to the pointillistic nature of these images. The co localization analysis used here was performed using the Coloc-Tesseler software, a method specifically designed for SMLM data. It exploits the space partitioning capability provided by Voronoi diagrams to compute Manders and Spearman's coefficient (Levet & al, *Nat Commun* **10** (2019)). The method is now widely applied in the field. In addition, as internal controls, we have validated our approach using experimental positive co-localization and artificially created positive and negative co-localization data. However, following the reviewer's question, we nevertheless tested CBC on several of our data sets: the results were not significantly different, as shown on the example below for CRT/CD47 and CRT/ERP57 co localizations:

* Apoptotic cells tend to be fragmented and scatter broadly the excitation light. The treatment of free aldehydes is appreciated. Still, the authors should show that the apoptotic cells do not show such scattering in unlabeled cells, as compared to labeled cells. This control is needed in each of the spectral regions used for fluorescence detection.

It is true that apoptotic cells tend to broadly scatter the excitation light, inducing higher background signal. We tested the level of background obtained on our sample in the absence of labeling and optimized our protocol to minimize it, in particular by washing cells with 100 μ M CuSO₄ in ammonium acetate (as precised in the materials and methods line). In such conditions, the SMLM data collected on blebbing apoptotic cells were of similar quality as those collected on viable cells.

Fig. 1 - The authors should also use non-specific markers (not just ERp57) as negative controls. A non-specific marker of membrane could be useful.

It is hard to predict which signal might be systematically and unambiguously non-localized with CRT. Using a general membrane marker would require a different protocol than those applied for cell surface labeling of macromolecules, and would produce a largely continuous labeling throughout the membrane, a situation not easily comparable to that of two more

sparingly distributed proteins. This is the reason why we have chosen to produce an artificial negative control, nevertheless based upon a real image acquisition as shown in fig S4. This seems to us the most appropriate way to evaluate the Spearman value from two labels that are not specifically co-localized, in experimental conditions identical to those used in our study.

Fig. 4 - What is the origin of the specific band at 180kDa? How are the non-specific bands at 34kDa explained?

The origin of the band at 34 kDa is unknown but it is present also on the control lane, indicating its non-specificity, likely related to our immunoprecipitation conditions.

The band at 180 kDa possibly corresponds to remaining aggregates between the rabbit antibody against integrin $\beta 3$ subunit and CD47 as this band was also revealed with an anti-rabbit IgG antibody.

Fig. 4 - The identification of restricted areas as microvilli or vesicle domains is not convincing. These domains could be caveolae or other PM-related structures. The authors are advised to use non-specific PM markers or other more specific markers (actin) to understand the nature of these domains. Can the authors employ 3D STORM to better visualize these structures?

We agree that the nature of the restricted areas where CD47 is sequestered are not fully characterized. This is why we remain cautious in writing that they are “possibly assigned to microvilli and vesicle-like domains”. Regarding Caveolae, their size (diameter of about 60 nm, PNAS, 113 (50) 14170-14172; <https://doi.org/10.1073/pnas.1617954113>) is not in agreement with the structures we observed, which are significantly larger. To start addressing the reviewer’s concern and characterize the microvilli-like structures, we have used TopFluor PS, a fluorescent analogue of phosphatidylserine that incorporates the membrane and thus could co-stain these structures. As shown below, the fact that this staining reveals the same structures as those seen in CD47 Single Particle tracking, is in favor of microvilli. Nevertheless we feel that adding extensive data to the manuscript to confirm this hypothesis would go beyond its scope and is not needed to reach our conclusions.

PS and CD47 distribution analysis. HeLa cells were incubated for 10 min with TopFluor PS (10 μ M) and CD47 was detected thanks to B6H12 antibody as described in Materials and Methods (single particle tracking paragraph). Images were obtained in TIRF mode. First,

TopFluor image was acquired (left panel) and then CD47/AF647 localizations from 4000 images were compiled (right panel).

Fig. 6 - I suggest that the authors specify how the different interactions depend on the perturbations? (i.e. instead of description via 'dependent', specify if the interaction is actually elevated or diminished)

We have modified the scheme in new Figure 8 accordingly.

Additional information: Due to addition of new images in the initial Figure 5, we have split its content into three figures (Figures 5-7):

The new Figure 5, corresponding to the panels a, b and c of the previous Figure 5.

Figure 6, contains the images.

Figure 7 contains the data previously presented in Figure 5 (e and f).

Figure 8 corresponds to the previous Figure 6

Reviewers' comments:

Reviewer #1 (Remarks to the Author):

The paper is improved and the authors have addressed my primary concerns.

Reviewer #2 (Remarks to the Author):

This reviewer has no more question

Reviewer #3 (Remarks to the Author):

In general, the authors have properly addressed my comments. Still, a significantly larger number of analyzed cells related to the STORM data (where currently $n=3$) is highly recommended.

Reviewer 3

in general, the authors have properly addressed my comments. Still, a significantly larger number of analyzed cells related to the STORM data (where currently $n=3$) is highly recommended.

In response, we have done new series of image acquisition for all of the different double labelling experiments. The new figure 1e includes 10 cells per conditions and statistical analysis comparing the Spearman's rank correlation values was added. Analysis of co-localization on selected areas (former figure 1e) is now added as supplemental figS4d.